# Population-scale long-read sequencing uncovers transposable elements associated with gene expression variation and adaptive signatures in *Drosophila*

Gabriel E. Rech[1], Santiago Radío[1], Sara Guirao-Rico[1], Laura Aguilera [1], Vivien Horvath[1], Llewellyn Green[1], Hannah Lindstadt[1], Véronique Jamilloux[2], Hadi Quesneville[2] & Josefa González [1✉]

High quality reference genomes are crucial to understanding genome function, structure and evolution. The availability of reference genomes has allowed us to start inferring the role of genetic variation in biology, disease, and biodiversity conservation. However, analyses across organisms demonstrate that a single reference genome is not enough to capture the global genetic diversity present in populations. In this work, we generate 32 high-quality reference genomes for the well-known model species *D. melanogaster* and focus on the identification and analysis of transposable element variation as they are the most common type of structural variant. We show that integrating the genetic variation across natural populations from five climatic regions increases the number of detected insertions by 58%. Moreover, 26% to 57% of the insertions identified using long-reads were missed by short-reads methods. We also identify hundreds of transposable elements associated with gene expression variation and new TE variants likely to contribute to adaptive evolution in this species. Our results highlight the importance of incorporating the genetic variation present in natural populations to genomic studies, which is essential if we are to understand how genomes function and evolve.

[1] Institute of Evolutionary Biology (CSIC-Universitat Pompeu Fabra), 08003 Barcelona, Spain. [2] Université Paris-Saclay, INRAE, URGI, 78026 Versailles, France. ✉email: josefa.gonzalez@ibe.upf-csic.es

Despite their crucial role and high prevalence in most eukaryotic genomes, transposable elements (TEs) and other structural variants (SVs) remain largely under-studied. This is mainly a consequence of the limitations of high throughput sequencing read length, tightly restricted to short-reads in the last decades[1–3]. Short-reads not only limited the annotation of SVs to what inference methods were able to identify[4–9], but also required a reference genome to map the reads, which has at least three major drawbacks: (i) the information about the genetic background and genomic context of the SVs are usually lost[4]; (ii) the analyses are biased to what is possible to identify using a specific reference genome[3,10,11]; and (iii) repetitive sequences in the reference genome are not well characterized when they are longer than the sequenced reads[12]. In the particular case of TEs, the limitations of using short-reads are exacerbated even further for two reasons: sequence divergence of the copies, and their extremely repetitive nature[13]. Such a complexity has severely restricted inter- and intra-species TE dynamics studies, a crucial aspect that needs to be addressed in order to better understand the organization, function, and evolution of genomes[14].

During the last years, technological developments in DNA sequencing read length have lead not only to an improvement in the quality and completeness of reference genomes[15–20], but also to a significant rise in the number of high-quality genomes for multiple individuals of the same species, opening a new era in comparative population genomics[21,22]. The ability of long-reads to span repetitive regions of the genome, together with the relative low price of generating sequences for several individuals, has opened up the possibility of resolving and comparing previously absent or misassembled regions in the genome[3,8,23–25], which can lead to a significant improvement in our ability to study TE structure, activity and dynamics in different organisms[20,26,27].

*Drosophila melanogaster* represents one of the best model animals for studying TEs, not only for having one of the best annotated eukaryotic genomes[28,29], but also for containing several active TE families[30]. Interestingly, even in such a well-studied organism, long-read sequencing approaches have made novel insights into the evolutionary dynamics of TEs[8,31,32]. However, these studies do not take full advantage of the variability present in the populations analyzed, as they mainly use standard homology-based approaches (e.g., *RepeatMasker* and *RepBase*) for annotating and analyzing TEs, which limits their analysis to TE families already present in the available libraries.

Here, we used long-read sequences to generate high quality genome assemblies for 32 *D. melanogaster* natural strains collected mainly in Europe from populations located in five different climatic regions and belonging to three of the five main climate types (Fig. 1). We used this new genomic resource for the de novo construction and manual curation of a library of consensus TE sequences that account for the variability observed in natural populations. Genome annotations performed with this manually curated library of TEs not only outperformed the current *D. melanogaster* gold-standard TE annotation (FlyBase), but also showed significant improvements compared with the state-of-the-art short-read-based methods for TE annotation. Furthermore, a joint in-depth analysis of TE copies annotated in the 32 newly sequenced genomes, 14 additional worldwide high-quality genomes, and the reference genome, revealed that analyzing 20 genomes is sufficient to recover most of the common genetic variation in out-of-Africa *D. melanogaster* natural populations; identified hundreds of TEs associated with changes in expression of their nearby genes; and allowed to identify 31% more TEs with evidence of positive selection compared with the previous most extensive analysis[33].

## Results

### Thirty-two highly complete *D. melanogaster* genomes in terms of genes and transposable elements

In order to access as much TE diversity as possible in natural populations of *D. melanogaster*, we performed sequencing and de novo genome assembly of 32 strains using long-read sequencing technologies (Fig. 1, Table 1, Supplementary Data 1 and 2, Supplementary Note 1). These 32 strains were collected from 12 geographical locations: 24 strains were collected from 11 European locations and eight strains were collected in a North American population[34]. These 12 populations represent five different climatic regions belonging to three main climatic types: arid, temperate, and cold (Fig. 1; Supplementary Data 1). Long-read sequencing resulted in 458.7 Gb, representing a theoretical average coverage of 82X (ranging from 45X to 123X) and average read length > 5.6 Kb, which has been previously shown to be sufficient for generating highly contiguous genome assemblies in other Drosophila species[35]; Supplementary Data 2).

Genome assembly, polishing, deduplication and contaminant removal resulted in genomes with a number of contigs ranging from 153 to 1185 (average 367), genome sizes from 136.6 Mb to 151.3 Mb (average 142 Mb), N50 values from 400 Kb to 18.9 Mb (average 3.8 Mb) complete BUSCO scores between 96.1% and 99%, and per base quality values (QV scores) between 37.2 and 52.9 (Table 1 and Supplementary Notes 2–4). CUSCO scores, i.e., percentage of contiguously assembled piRNA clusters[36], range from 35.3% to 84.7% (average 64.1%; Table 1). The detectability of a cluster was inversely correlated with its size (Pearson´s correlation = −0.47; Supplementary Data 3b, Supplementary Fig. 1 and Supplementary Note 5). Although the high variability, these results are comparable with genomes previously obtained using similar sequencing and assembling strategies[35]. Note that differences in sequencing coverage did not explain the observed differences in genome size or TE content across genomes (Supplementary Fig. 2). Similarly, differences in read length and N50 values do not correlate with differences in genome size, TE content, or BUSCO scores (Supplementary Fig. 2).

After reference-guided scaffolding using the ISO1 reference genome, on average >90% of the contigs mapped to major chromosomal arms, which contained >98.5% of the bases in the

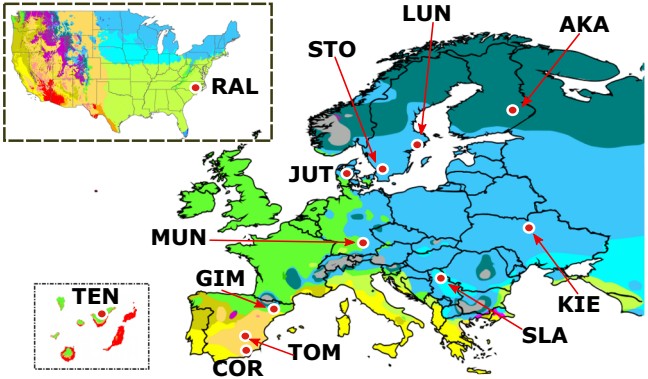

**Fig. 1 Geographical location of the 12 *D. melanogaster* natural populations analyzed in this work.** The 32 sequenced and assembled genomes correspond to strains obtained from: Tenerife, Spain: TEN (1), Munich, Germany: MUN (6), Gimenells, Spain: GIM (2), Raleigh, USA: RAL (8), Cortes de Baza, Spain: COR (4), Tomelloso, Spain: TOM (2), Jutland, Denmark: JUT (2), Stockholm, Sweden: STO (1), Lund, Sweden: LUN (2), Slankamen, Serbia: SLA (1), Kiev, Ukraine: KIE (1) and Akka, Finland: AKA (2). In brackets, the number of genomes sequenced from each location. Map colors represent different climatic regions according to the Köppen climate classification (Supplementary Data 1).

**Table 1 Summary of assembly metrics of the 32 genomes sequenced in this work.**

| Strain | Location | Contigs | Genome size | N50 (Mb) | BUSCO complete | BUSCO duplicate | QV | c.CUSCO | sc.CUSCO | Completeness (ISO1 aligned bases) | Euchromatic size (Mb) |
|---|---|---|---|---|---|---|---|---|---|---|---|
| AKA-017[a,b] | Akka, Finland | 164 | 142.7 | 18.9 | 98.7% | 0.50% | 51.04 | 82.35% | 94.12% | 96.30% | 100.1 |
| AKA-018[c] | Akka, Finland | 162 | 136.7 | 2.3 | 98.4% | 0.70% | 37.63 | 72.94% | 92.94% | 93.50% | 100.9 |
| COR-014[a,b] | Cortes de Baza, Spain | 161 | 138.1 | 7.7 | 98.3% | 0.50% | 43.62 | 72.94% | 96.47% | 96.70% | 100.4 |
| COR-018[c] | Cortes de Baza, Spain | 402 | 143.5 | 0.9 | 98.0% | 1.00% | 38.47 | 55.29% | 96.47% | 94.30% | 103.3 |
| COR-023[c] | Cortes de Baza, Spain | 620 | 139.5 | 0.6 | 97.8% | 0.80% | 37.42 | 35.29% | 92.94% | 93.60% | 101.5 |
| COR-025[c] | Cortes de Baza, Spain | 377 | 143.4 | 0.7 | 98.1% | 1.00% | 37.83 | 57.65% | 92.94% | 94.00% | 102.7 |
| GIM-012[c] | Gimenells, Spain | 383 | 140 | 1.2 | 98.4% | 0.80% | 40.56 | 45.88% | 87.06% | 94.10% | 101.2 |
| GIM-024[a,b,c] | Gimenells, Spain | 316 | 142.3 | 6.8 | 99.0% | 0.50% | 50.77 | 77.65% | 94.12% | 95.20% | 100.2 |
| JUT-008[c] | Jutland, Denmark | 330 | 148.5 | 9.6 | 98.4% | 0.50% | 49.52 | 80.00% | 96.47% | 93.60% | 101.5 |
| JUT-011[a,b] | Jutland, Denmark | 184 | 138.4 | 4 | 98.7% | 0.50% | 44.94 | 70.59% | 98.82% | 96.50% | 100.8 |
| KIE-094[a,b] | Kiev, Ucrania | 343 | 143.8 | 3.8 | 98.7% | 0.80% | 48.78 | 75.29% | 96.47% | 96.20% | 101.9 |
| LUN-004[a,b] | Lund, Sweden | 314 | 138.1 | 2 | 98.7% | 0.60% | 44.24 | 62.35% | 96.47% | 96.30% | 101.1 |
| LUN-007[c] | Lund, Sweden | 360 | 142.4 | 1.1 | 98.0% | 0.60% | 39.91 | 52.94% | 95.29% | 94.10% | 102.1 |
| MUN-008[c] | Munich, Germany | 250 | 142.2 | 1.1 | 97.5% | 0.90% | 37.76 | 68.24% | 94.12% | 94.10% | 101.7 |
| MUN-009 | Munich, Germany | 385 | 149.3 | 5.6 | 97.9% | 0.50% | 45.97 | 71.76% | 95.29% | 94.10% | 102.1 |
| MUN-013[c] | Munich, Germany | 406 | 138.4 | 1 | 98.2% | 0.50% | 39.28 | 49.41% | 90.59% | 93.80% | 101.9 |
| MUN-015 | Munich, Germany | 251 | 140 | 1.2 | 98.0% | 1.00% | 38.19 | 65.88% | 92.94% | 93.90% | 101.8 |
| MUN-016[a] | Munich, Germany | 217 | 142 | 7.8 | 98.50% | 0.60% | NA | 77.65% | 92.94% | 96.60% | 100.7 |
| MUN-020[c] | Munich, Germany | 324 | 138.1 | 1.3 | 97.10% | 1.10% | 40.93 | 48.24% | 82.35% | 93.80% | 101.2 |
| RAL-059[c] | Raleigh, USA | 688 | 143.5 | 0.8 | 98.10% | 0.90% | 43.25 | 51.76% | 94.12% | 93.20% | 101.7 |
| RAL-091[c] | Raleigh, USA | 887 | 145.1 | 0.5 | 97.50% | 1.00% | 44.04 | 57.65% | 92.94% | 92.80% | 103.9 |
| RAL-176[c] | Raleigh, USA | 1185 | 151.3 | 0.4 | 97.10% | 0.80% | 46.62 | 53.53% | 88.24% | 92.70% | 102.9 |
| RAL-177[a,b,c] | Raleigh, USA | 188 | 141.9 | 14.6 | 97.40% | 0.40% | 46.70 | 84.71% | 96.47% | 95.70% | 100.7 |
| RAL-375[a,b,c] | Raleigh, USA | 179 | 141.2 | 13.5 | 96.10% | 0.40% | 44.86 | 82.35% | 96.47% | 96.10% | 100.7 |
| RAL-426[c] | Raleigh, USA | 500 | 137 | 0.7 | 97.60% | 0.50% | 38.04 | 51.76% | 90.59% | 93.50% | 102.0 |
| RAL-737[c] | Raleigh, USA | 469 | 147.8 | 1.5 | 97.40% | 0.50% | 42.11 | 70.59% | 95.29% | 93.20% | 102.1 |
| RAL-855[c] | Raleigh, USA | 332 | 144.4 | 3.9 | 97.00% | 0.40% | 41.78 | 78.82% | 97.65% | 93.40% | 102.2 |
| SLA-001[a,b] | Slankamen, Serbia | 432 | 143.7 | 0.8 | 97.90% | 0.80% | 38.45 | 58.82% | 97.65% | 96.60% | 103.0 |
| STO-022[a,b] | Stockholm, Sweden | 153 | 142.4 | 3.1 | 98.10% | 0.70% | 36.00 | 71.76% | 96.47% | 96.90% | 102.5 |
| TEN-015[a,b] | Tenerife, Spain | 329 | 140.5 | 1.1 | 97.90% | 1.00% | 40.30 | 61.18% | 94.12% | 96.20% | 102.0 |
| TOM-007 | Tomelloso, Spain | 222 | 139.5 | 3.2 | 98.20% | 0.70% | NA | 57.65% | 92.94% | 96.90% | 101.0 |
| TOM-008[a,c] | Tomelloso, Spain | 219 | 136.6 | 1.9 | 98.10% | 0.80% | 41.75 | 61.18% | 85.88% | 94.10% | 101.3 |
| ISO1-Sol[d] | Reference Genome | 518 | 147.8 | 3.4 | 96.00% | 0.50% | 42.92 | 77.65% | 91.76% | 97.57% | 101.9 |

Genomes were sequenced using ONT except MUN-016 and TOM-007 that were sequenced using PacBio.
Additional information on the strains can be found in Supplementary Data 1 and on the sequencing in Supplementary Tables S2 and S3. Besides contig-CUSCO scores (c.CUSCO) scaffold-CUSCO scores (sc.CUSCO) are also given (the later values are higher as expected if the piRNA flanking regions are present in the assembled genomes).
[a]The 13 strains used in the construction of the de novo MCTE library.
[b]The 11 strains used in the comparison of TE annotations using *REPET, TIDAL* and *TEMP.*
[c]The 20 strains used in the cis-eQTL analysis.
[d]Genome assembled using long-read sequencing data of the *D. melanogaster* reference genome provided in Solares et al.[18].

de novo assembled genomes (Supplementary Data 3a). The scaffolded genomes also showed a high level of completeness, covering on average around 95% of ISO1 major chromosomal arms (Table 1, Supplementary Fig. 3) and with an average of 99.75% of the protein coding genes successfully transferred (Supplementary Data 3a).

To quantify the accuracy of the TE sequences generated with long-read sequencing, we used our pipeline (from base calling to genome scaffolding) to process the ONT long-reads available for the reference genome[18]. The newly assembled reference genome was 147.8 Mb with a complete BUSCO score of 96% (Table 1). We identified 1842 orthologous TE insertions between our assembly and the FlyBase reference genome, with 99.9% pairwise identity suggesting that our pipeline produces highly accurate TE sequences (Supplementary Data 4). We also used the pipeline applied in Berlin et al.[37] to annotate TEs in an ISO-1 assembly based on PacBio sequencing, to annotate TEs in our ISO-1 assembly based on the Solares et al.[18] ONT reads. We found that 18% more TE insertions were annotated when using the Berlin et al.[37] assembly, suggesting that besides TE annotation pipelines, sequencing and assembly strategies can also influence the annotation of TEs in genomes.

Overall, we generated 32 de novo D. melanogaster assembled genomes from 12 geographically diverse populations that are contiguous and complete in terms of gene and TE content.

**A new manually curated library of consensus sequences allowed the annotation of 58% more TE copies in the high-quality _D. melanogaster_ reference genome.** In order to accurately annotate TE copies in the 32 de novo assembled genomes of _D. melanogaster_, we implemented a TE annotation strategy involving, as a first step, the generation of a manually curated TE (MCTE) library. The MCTE library was built using the _REPET TEdenovo_ pipeline for the de novo prediction of consensus sequences representative of TE families[38]. Because the library required extensive manual curation, we focused on 13 genomes that represent the 12 geographical locations in our analysis (Table 1). Overall, the _TEdenovo_ pipeline reconstructed 28,009 consensus sequences. After manual curation (Supplementary Note 6), the MCTE library ended up with 165 consensus sequences, which are 34 more sequences than the ones present in the Berkeley Drosophila Genome Project (BDGP) dataset for _D. melanogaster_[39] (Supplementary Data 5). The MCTE library sequences are representative of 146 TE families (13 of them represented by more than one consensus sequence), including three new families (see below).

The second step of the annotation process used the _TEannot_ pipeline of _REPET_ to annotate all the TEs present in each one of the 32 genomes and the reference ISO1 genome using the MCTE library. The euchromatic region analyzed ranged from 100.1 Mb to 103.9 Mb (Table 1), which is a slightly larger region than in previous similar analysis (e.g., 94.5 Mb in Charkraborty et al.[8]). As a proof of concept, we compared the euchromatic TE annotation performed with _REPET_ with the current TE annotation available in FlyBase, which is considered the gold-standard[28]. We found that all but two families in FlyBase were present in the _REPET_ annotations: _frogger_ and _gypsy3_, with only one copy each annotated in FlyBase. _REPET_ most likely fails to detect the _frogger_ copy because it is nested in a _copia1_ insertion, while the only copy of _gypsy3_ is annotated in the heterochromatin and thus not included in our _REPET_ annotations. When considering only those families present in both annotations, we observed no significant differences in the number of copies between REPET and FlyBase annotations (FDR $p$-value >0.05, $X^2$ test, Fig. 2a, Supplementary Data 6a), with the exception of the INE-1 elements, for which REPET annotated a larger number of

copies than FlyBase (FDR $p$-value <0.0001, $X^2$ test, Supplementary Data 6a). At the genomic coordinates level, ~85% of the FlyBase copies were overlapping with _REPET_ copies (95% reciprocal minimum breadth of coverage; Fig. 2b, Supplementary Data 6b). Moreover, overall sensitivity and specificity of _REPET_ annotation when comparing with FlyBase were 99.44% and 99.29%, respectively (calculated according to Quesneville et al.[40]; Supplementary Data 6c). Thus, overall the annotation of the reference genome performed with the MCTE library was able to reproduce with high accuracy the FlyBase TE annotation, the current gold-standard TE annotation in _D. melanogaster_[29].

However, while the number of copies and the coordinates of TEs from families present in both annotations were very similar, our annotation strategy allowed us to annotate 468 copies from 28 TE families not present in the FlyBase annotation. While most of them correspond to known TE families, such as _LARD_, _Kepler_ and _THARE_, 27 copies correspond to three new TE families (see below). Moreover, 15 copies belong to families such as _gypsy10_, _BS4_ and _ZAM_, which according to FlyBase were only present in the heterochromatic regions, but we found them in euchromatic regions as well (Supplementary Data 6a, Fig. 2a). Although most of the new TE copies annotated only with _REPET_ were small insertions, we also identified 50 insertions larger than 2 Kb (Fig. 2c, Supplementary Note 7).

We further compared the number of TEs annotated in the 13 genomes with the previously available _D. melanogaster_ BDGP library and with the MCTE library (Supplementary Data 6d, Supplementary Note 7). We found that 42–44% of the copies annotated using the MCTE library were not annotated by the BDGP library.

Overall, by creating a library that contains the TE diversity of 13 _D. melanogaster_ strains from 12 geographical locations, we were able to identify TE copies from 25 known families not previously annotated in the reference euchromatic genome, and from three new families (see below). In total, 58% more insertions were annotated in the euchromatic reference genome using the MCTE library (1301 FlyBase vs 2059 _REPET_), and 42–44% more copies were identified using the MCTE library compared with the BDGP library when analyzing 13 other genomes.

**The new manually curated TE library allowed the identification of three new families in _D. melanogaster_, two of them also present in other Drosophila species.** Three consensus sequences in the MCTE library that failed to be assigned to any known family in the BDGP or the _RepBase_ database were further analyzed using _PASTEC_[41]. These new consensus sequences were classified as a Miniature Inverted Repeat Transposable Element (_MITE_), a Terminal Repeat Retrotransposon in Miniature (_TRIM_), and a Terminal Inverted Repeat (_TIR_) element (Fig. 3a).

Numerous Bari-like _MITEs_[42] and Mariner-like _MITEs_[43] have been previously described in _D. melanogaster_. However, the _MITE_ consensus sequence identified in this work showed no significant alignments with any previously described _MITEs_ (nucleotide identity percentage <50%), suggesting that it belongs to a new undescribed _MITE_ family. On average, more than eight _MITE_ copies were found in each _D. melanogaster_ strain. Identified copies were of variable length (Supplementary Fig. 4a) and highly similar (average identity >89%, Supplementary Fig. 5). Moreover, the consensus sequence of the new _MITE_ family showed no significant similarities with TEs identified in other five Drosophila genomes (Supplementary Data 7, Supplementary Note 8), suggesting that this element could have invaded the _D. melanogaster_ genome recently.

Regarding the new _TRIM_ element, while the consensus sequence showed the typical _TRIM_ structure (less than 1000 bp, with LTRs sequences between 100 bp and 250 bp, Fig. 3a), no

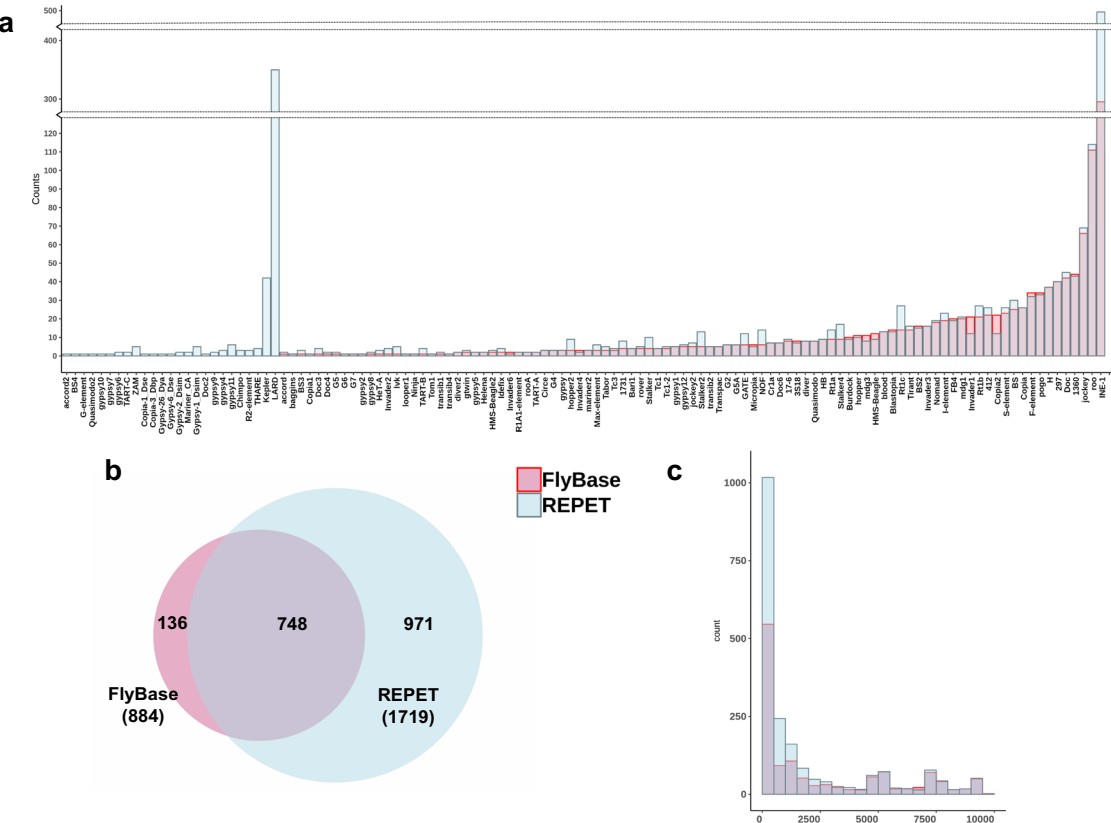

**Fig. 2 Comparison between the TE annotation in FlyBase and the TE annotation performed using _REPET_ with the MCTE library.** In blue copies annotated with REPET and in red copies annotated by FlyBase in the reference genome. **a** Number of TE copies per family. **b** Overlapping of TE annotations considering that the copies were from the same family and that they were overlapping at least 95% of their lengths (breadth of coverage). TEs shorter than 100 bp, belonging to the INE-1 family and nested TEs were excluded from the analysis. **c** Distribution of number of TE copies by length in 500 bp bin sizes.

similarities with any known TE in public databases was found. Notably, this sequence was not the only _TRIM_ element in the MCTE library since other _TRIM_ consensus sequence showing similarities with a _Kepler_ element was also found. Most copies of the new _TRIM_ element have the size of the consensus sequence (Fig. 3b), however we found relatively low similarity among the copies (average identity 77%, Supplementary Fig. 5) and evidences that the element is present in at least another Drosophila species (_D. pseudoobscura_, Supplementary Data 7, Supplementary Note 8), suggesting that this _TRIM_ element could represent the remains of an ancestral TE family.

Finally, the newly identified _TIR_ element showed 51% sequence similarity to the internal domain of _EnSpm-1_JC_, a _TIR_ element from the _Jatropha curcas_ genome[44] (Fig. 3a and Supplementary Fig. 4b). Moreover, while the consensus sequence did not actually contain the inverted repeats at the ends (_TIRs_), we found 31%-43% of the copies annotated in each of the 32 genomes to contain degraded inverted repeats in the 1 kb flanking regions (Supplementary Note 9). Besides, average copy identity per genome was low (68%, Supplementary Fig. 5) and most copies were truncated representations of the consensus (Fig. 3b). These results, coupled with the similarity showed by the new _TIR_ element against TE consensus sequences from _D. virilis_ and _D. bipectinata_ (Supplementary Data 7, Supplementary Note 8), suggested an ancient origin for this element.

Thus, even in a well-studied species as _D. melanogaster_, the de novo TE annotation and manual curation using a long-read strategy in a geographically diverse panel of strains allowed the identification of three new TE families. Copies from two of these families (_TRIM_ and _TIR_ elements) showed low levels of similarity suggesting that

they are old insertions; while copies of the new _MITE_ family were highly similar suggesting that it might have recently transposed.

**Short-read methods failed to detect up to 57% of the insertions detected by long-read based annotation.** Besides comparing our TE annotations with those available in FlyBase, we also wanted to investigate how de novo annotations based on long-read sequencing assemblies compare with annotations based on short-read sequencing. Previous estimates suggested that short-reads failed to find 36–38% of the TE insertions annotated based on long-reads[8,9]. To estimate this percentage in our genomes, we compared the results obtained with the MCTE library in long-reads using _REPET_, and in short-reads using two different tools: _TEMP_[45] and _TIDAL_[46] (Supplementary Data 8). For this comparison, we focused on 11 of the most complete genomes representative of the geographic variability of our samples and included in the previous subset of 13 genomes used to build the MCTE library (Table 1, Supplementary Note 10).

The total number of TE insertions detected by each software was more similar for _REPET_ and _TEMP_ (6632 and 7430, respectively) than for _TIDAL_ (9066) (Supplementary Data 8a). The number of TE insertions detected both by _REPET_ and _TIDAL_ (4041) is higher that the number of TE insertions detected by _REPET_ and _TEMP_ (3254). The overlap of the insertions detected both by _TIDAL_ and _TEMP_ is higher (4786), probably because the methodologies of these two software are more similar (Supplementary Data 8a).

To estimate the false negative rate of _TEMP_ and _TIDAL_ and the false positive rate of _REPET_, we performed manual inspection

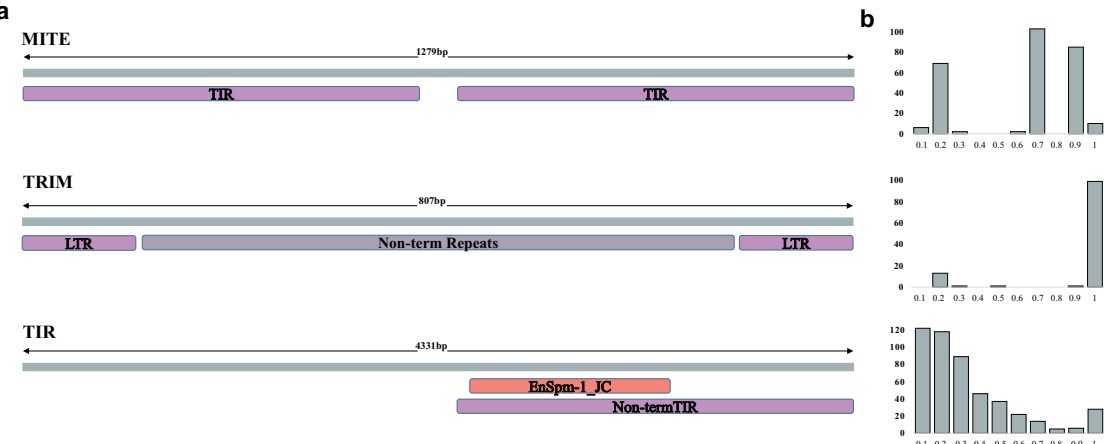

**Fig. 3 Three new TE families in *D. melanogaster*. a** Schematic representation of the structural features detected by *PASTEC* in the consensus sequences of the three new families identified in this study. **b** Length ratio (size as proportion of the consensus) distribution for TE copies annotated in the 32 genomes with each of the three new consensus sequences.

for 300 TE insertions annotated by *REPET*. When comparing the TE annotations between *REPET* and *TEMP*, 120 TEs (40%) were correctly annotated by the two software, while 170 (57%) TEs annotated by REPET were missed by *TEMP* (Supplementary Data 8b). When comparing *REPET* and *TIDAL* annotations, 212 TEs (71%) were correctly annotated by the two software, while 78 TEs (26%) were correctly annotated by REPET and missed by TIDAL (Supplementary Data 8b). Finally, 10 of the 300 TEs annotated by REPET, were false positives as we could not confirm their presence using Blast (see Methods).

Additionally, we performed manual inspection of 50 TEs that were identified by *TEMP/TIDAL* but were not identified by *REPET* (Supplementary Data 8c). None of these insertions were present in the genome assemblies. For these TEs, we could not distinguish whether they were *REPET* false negatives or *TEMP/TIDAL* false positives. However, the majority of these insertions (39/50) have a frequency estimate <20% according to *TEMP*, suggesting that they could be false positives[45]. For the 11 TEs with frequencies >20% we cannot discard that these correspond to *REPET* false negatives as *REPET* is run on the assembled genomes that contain a single haplotype, while software based on short-reads allow the interrogation of all the haplotypes present in a given sample (Supplementary Data 8c).

Thus overall and depending on the tool, short-read tools fail to annotate 26–57% of the TEs annotated using long-read tools, while *REPET* false positive rate was 3%.

**TE content is similar across *D. melanogaster* strains while TE activity varies**. When comparing TE annotations for the 32 genomes plus the reference genome (ISO1), we observed low variation among strains regarding both TE content (percentage of the euchromatic genome occupied by TEs, average = 3.56%, SD = 0.3%) and number of TE copies (average = 2016, SD = 69.6) (Supplementary Data 9a). The coefficient of variation for the number of non-reference insertions across populations was similar to previous estimates (7% vs 9% in Chakraborty et al.[8]). As previously described, TE variation across populations did not reflect the geographical or environmental origin of the populations[30] (Fig. 4a; see Methods).

At the TE order level, and in agreement with previous studies[30], we found *LTRs* to be the most abundant, representing near 60% of all TE content (Supplementary Data 9b, Supplementary Fig. 6a), while the number of TE copies was more evenly distributed among the five main orders (*Helitrons*, *LARDs*, *LINEs*, *LTRs* and *TIRs*)

(Supplementary Data 9b, Supplementary Fig. 6b). Also in agreement with previous observations, *INE-1* superfamily showed the largest number of copies among Class II DNA elements[47] and *Gypsy* and *Pao* elements were the most abundant among the LTRs[30,48] (Supplementary Data 9c). Moreover, while no overall significant differences in abundance were found at the superfamily level (Pearson's $X^2$ test of independence = 575.44, *p*-value = 0.4987, Fig. 4b, Supplementary Data 9c), genome pairwise comparisons were significant for the MUN-009 and ISO1 pair of strains ($X^2$ test, adjusted *p*-value = 0.03, Fig. 4c), mainly due to the *P* superfamily overrepresentation in MUN-009 compared with the ISO1 genome (Fig. 4d). This observation was also confirmed by the analysis at the family level, where MUN-009 was found to contain 60 copies of the *P-element*, while this element is absent from the ISO1 genome[49] (Supplementary Fig. 7 and Supplementary Data 9d). *P-elements* were indeed among the most variable families in the 33 genomes (Supplementary Fig. 8, Supplementary Data 9d).

We used the percentage of sequence identity between individual TE copies and the family consensus sequence, as a proxy for the age of the insertions. As expected, we found *INE-1* and *LARD* elements to be the oldest superfamilies in all genomes[50,51], while copies of the *I*, *TcMar-pogo*, *Copia* and *Pogo* superfamilies showed the highest values of identity with the consensus, suggesting they are relatively young, as also previously described[30,52] (Fig. 4e and Supplementary Fig. 9). Moreover, some superfamilies showed a large variability in identity such as *R1*, *Jockey* and *Gypsy*, indicating that they contain both young and old members (Fig. 4e and Supplementary Fig. 9). Genome pairwise comparisons in the distribution of identity values per genome showed significant differences between some pairs of genomes (Supplementary Fig. 10a). Notably, such differences seem to be mainly caused by members of the *Jockey* and *Gypsy* superfamilies (Supplementary Fig. 10b).

Our results, together with previous studies in Drosophila populations, suggest a scenario in which while natural variation in TE abundance between populations exist, certain families tend to be either abundant or rare in most populations[30,46]. Moreover, while almost no significant differences were observed between genomes in the number of TE copies (Fig. 4c), we did find pairwise differences in the identity of the copies (Supplementary Fig. 10a), particularly among members of two superfamilies, *Jockey* and *Gypsy* (Fig. 4e; Supplementary Fig. 10b), suggesting a population specific behavior regarding TE activity as previously described in both European[30] and North American strains[53].

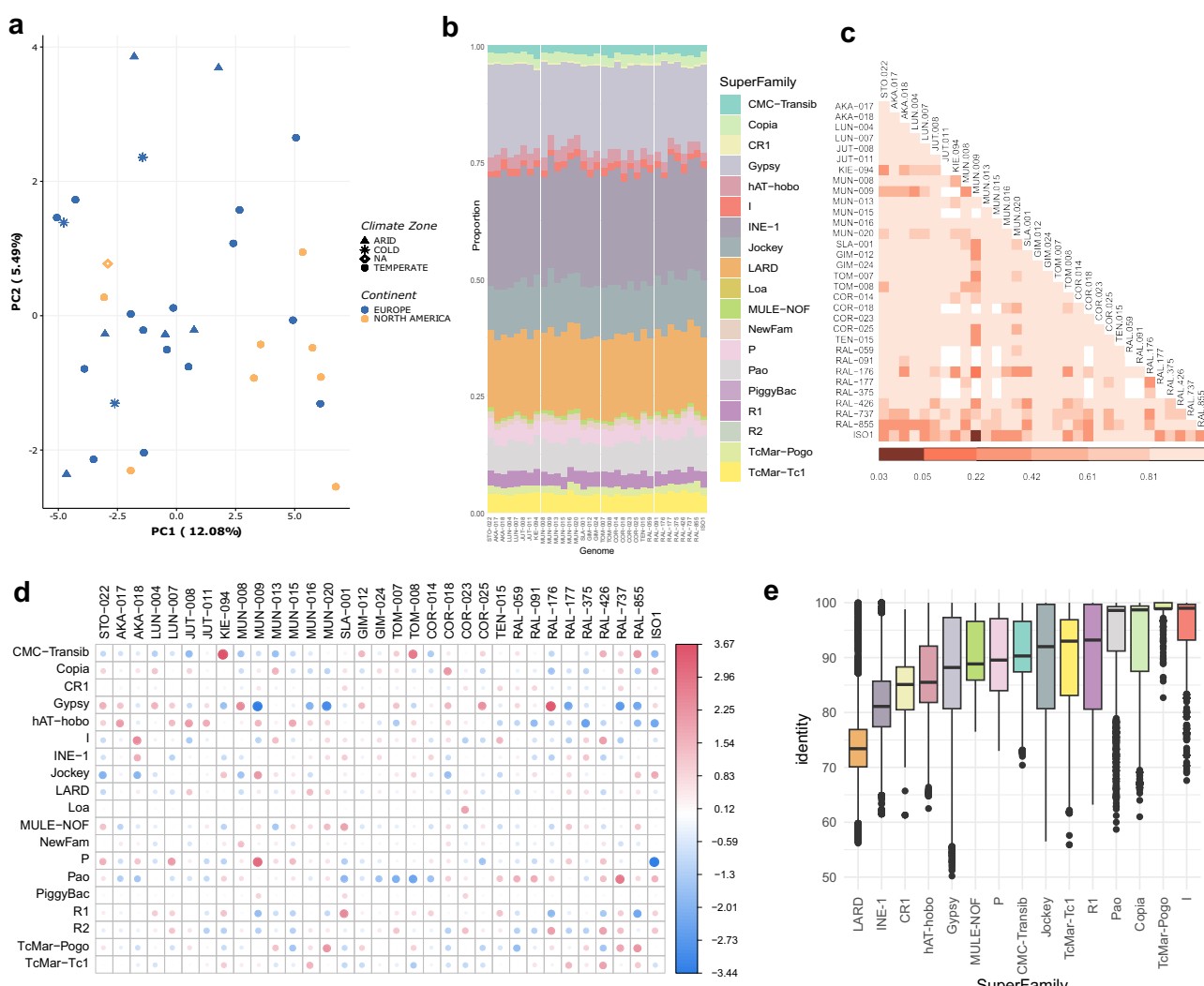

**Fig. 4 TE annotations at the superfamily level. a** Principal component analysis based on TE insertions polymorphisms grouped by continent (colors) and climatic zoned (shapes). **b** The proportion of TE copies annotated for each superfamily. **c** Per genome pairwise comparisons in the proportion of copies annotated at the superfamily level. The colors of the matrix squares represent adjusted (FDR) *p*-values of the two-sided Chi Square test. Only one significant result was observed (adjusted *p*-value = 0.03) between ISO1 and MUN-009. **d** Representation of the Pearson residuals (r) for each cell (pair Superfamily-genome). Cells with the highest residuals contribute the most to the total Chi Square score. Positive values in cells (red) represent more copies than the expected, while negative residuals (blue) represent fewer copies than the expected (does not imply statistical significance). **e** Distribution of TE insertion identity values classified by superfamily and considering all genomes together. The boxplot shows median (the horizontal line in the box), 1st and 3rd quartiles (lower and upper bounds of box, respectively), minimum and maximum (lower and upper whiskers, respectively). Number of copies analyzed per superfamily are given in Supplementary Data 9c.

**20 genomes allow the identification of the vast majority of TEs that are common in out-of-Africa natural populations.** To investigate how the number of genomes analyzed affects the total number of unique TE copies identified and the estimation of their population frequencies, we identified orthologous insertions by comparing the annotations obtained using *REPET* in 47 genomes: the 32 genomes sequenced in this work, the ISO1 reference genome, and the 14 genomes reported by Chakraborty et al.[8] collected in Africa (2), Europe (2), North America (4), North Atlantic Ocean (1), South America (2), and Asia (3) (Supplementary Data 10 and 11). On average, 2016 euchromatic TE copies were annotated per genome (ranging from 1883 to 2178, Supplementary Data 9a), and for 97% of them (on average) orthologous relationships of the insertion flanking regions in the ISO1 reference genome were determined (Supplementary Data 11a; Supplementary Note 11). Overall, we annotated 28,947 TEs across the 47 genomes (Supplementary Data 10). As

expected, the site frequency spectrum of TE insertions showed an excess of rare variants compared with SNP variants[54] (Supplementary Fig. 11).

We classified the 28,947 TEs in three frequency classes: rare (present in <10% of the genomes), common (present in ≥10% and ≤95%) and fixed (present in >95%) and calculated the number of TEs detected in each frequency class starting with the analysis of only five genomes and adding one genome at a time until the total 47 genomes available (see Methods). As expected, we found that as the number of genomes analyzed increased, the number of rare TEs also increased in a linear fashion, as each genome contributes a similar number of rare TEs to the population (Fig. 5a and Supplementary Data 11b). On the other hand, the number of fixed TEs was very similar regardless of the number of genomes considered, and the small variations seen were probably due to errors in either the TE transfer, TE annotation, or genome assemblies (Fig. 5a). Finally, we observed that the number of

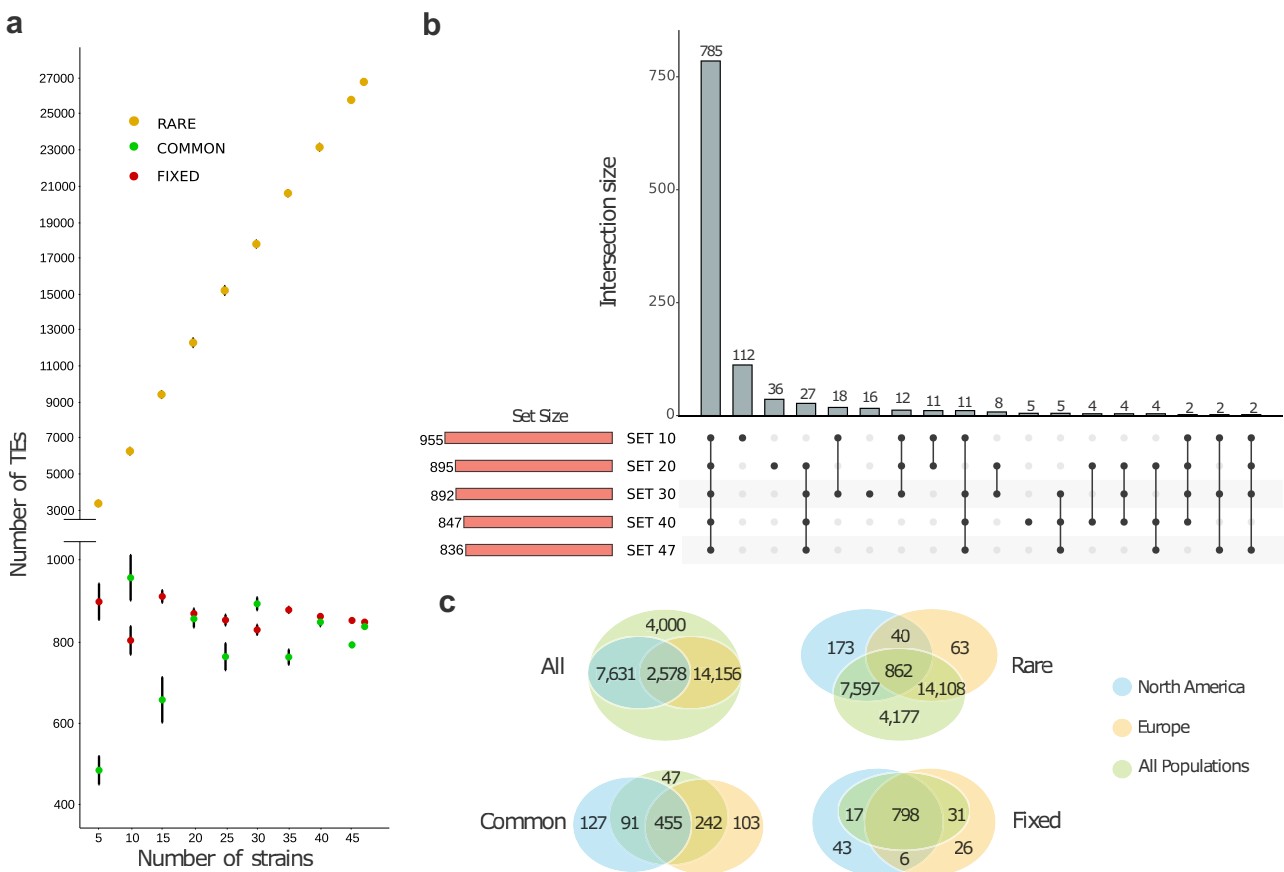

**Fig. 5 TE classification according to three frequency classes: rare (present in <10 of the strains), common (present in ≥10 and ≤95% of the strains) and fixed (present in >95% of the strains). a** Number of TEs and their classification according to their frequency in the population using from 5 to 47 strains. The standard deviation was calculated by taking 30 random samples of strains for each case. Data are presented as median values ±standard deviation. **b** Intersection of the different sets of common TEs identified taking into account 10, 20, 30, 40 and 47 strains at random. **c** Venn diagrams depicting the intersection of orthologous TEs defined by geographic origin. The ALL diagram represents all TEs regardless their frequency class, while the rare, common and fixed diagrams are defined by the TEs of each of the classes in each set.

common TEs is more variable depending on the number of genomes considered, and this number stabilizes around 800–900 TEs. The overlap of common TEs considering 10, 20, 30, 40 and 47 strains showed that most of the common TEs (785; 74%) were present in all the subsets (Fig. 5b). By increasing the number of genomes analyzed from 10 to 20, the number TEs identified as common decreased (Fig. 5b). Besides the core set of 785 common TEs detected in all the subsets, additional 112 TEs were detected as common when analyzing 10 genomes, while only 36 additional TEs were detected as common when analyzing 20 genomes, and 27 additional TEs when analyzing more than 20 strains (Fig. 5b). These results suggest that 20 genomes are enough to accurately identify most common TEs in populations, which is the subset of TEs expected to be enriched for candidate adaptive mutations[33].

To determine whether the geographical origin of the strains affects the total number of TE copies identified and their frequency classification, we analyzed genomes according to the continental origin of the sequenced strain: North America, Europe and All populations (Supplementary Data 11a). Most of the TE insertions were only identified in either Europe or North America (Fig. 5c). However, most of these were rare, reflecting the increase in the number of genomes analyzed rather than a geographical effect. On the other hand, if we focused on the common TE insertions, 127 insertions were unique to North America and 103 to Europe (Fig. 5c; Supplementary Data 11c). While some of these insertions were classified as fixed in the other continent, 70 of the common TEs only found in Europe were

absent in North America, while 47 of the common TEs found only in North America were absent in Europe (Supplementary Data 11d). These common TEs that are specific to a particular geographic region are good candidates to have a role in local adaptation. However, the number of TEs was too small to identify enriched biological processes in the genes nearby these TE insertions in these continents.

Overall, our results suggest that the analysis of 20 genomes accurately identifies most common and fixed TEs in a diverse set of populations. Still, because a proportion of the common TEs identified were continent specific, analyzing populations from other continents should lead to the identification of additional common TE insertions.

**Hundreds of de novo annotated TEs are associated with the expression of nearby genes**. To determine whether TE insertions were associated with the level of expression of nearby genes, we looked for significant associations between cis-eQTLs and TE insertions using RNA-Seq data available for 20 of the strains in our dataset[55–57] (Table 1, Supplementary Data 2c). We focused on TE insertions located in high recombination regions as those insertions are more likely to be causal mutations. We identified 503 significant associations (adjusted $p$-value <0.05), including 481 genes and 472 TEs, the majority of them annotated in this work for the first time (470; Supplementary Data 12a). Also, most of them (433 out of 472; 91.7%) were present at low frequencies

in populations (≤ 5%) suggesting that their effect on gene expression could be deleterious. These TEs were enriched for members of the *P* superfamily and for the *P-element*, *transib1*, *Gypsy-2_Dsim*, *412* and *Doc* families ($X^2$ test, *p*-value <0.05, Supplementary Data 12b). Genes located nearby these TEs were not significantly overrepresented for any biological process, molecular function or cellular component nor any metabolic pathways[58,59]. Contrary to previous results, we found a similar number of low frequent TEs associated with gene up- and down-regulation[54] (214 vs 258, respectively; Supplementary Data 12a; *Gypsy-2_sim*, *1360*, *Copia* and *Blood* were enriched only nearby up-regulated genes, while *transib1* and *Doc* were only enriched nearby down-regulated genes (Supplementary Data 12c–d).

We manually curated the TE annotations that showed an adjusted *p*-value <0.01, and we confirmed 13 significant associations involving 13 genes and 14 TEs, as the *Ten-a* gene had two nearby TEs in linkage disequilibrium that were identified as the top variants (Fig. 6 and Table 2; see Methods). Several of the 13 most significant genes are involved in response to stimulus and could be candidates to play a role in the adaptation to new environments (Table 2). For example, *Cyp6a17*, is involved in temperature preference behavior[60] and it is located within a genomic region harboring several insecticide resistance genes from the *cyp* family[61]. Manual curation of this region revealed that strains with the TE insertion also had a triplication of the *Cyp6a17* gene that could also contribute to the increased level of expression found in strains with the TE insertion. *Gr64a*, is a gustatory receptor gene required for the behavioral responses to multiple sugars (glucose, sucrose, and maltose)[62]. Furthermore, other genes may be important for their role in neurogenesis (*pde9*, *ppk*[63]) and synaptic organization (*Ten-a*, *dpr8*[64], Table 2).

**Most of the insertions with signatures of selection in their flanking regions were de novo annotated insertions.** In order to identify TEs likely to play a role in adaptation, we looked for evidence of positive selection in the TE flanking regions. We used SNPs alleles as a proxy to identify genomic regions undergoing selective sweeps and then we explored whether such a sweep was linked to a nearby TE insertion. We applied three haplotype-based statistics: iHS[65], iHH12[66,67] and nSL[68]. We defined a SNP to have a significant iHS, iHH12 or nSL values when, after normalizing by frequency and chromosome location, the normalized values were >95th percentile of the distribution of values for SNPs falling in neutral introns (see Methods). We then looked for candidate adaptive TE insertions in linkage disequilibrium with each significant SNP, and located <1 kb from the significant SNP (see Methods). We considered as candidate adaptive TEs those present at high population frequency and located in regions with recombination rates >0 (see Methods and Rech et al.[33]). Among the 746 candidate adaptive TEs, we found 19 TEs co-occurring with SNPs showing evidence of selective sweeps (Supplementary Data 13a). Among these 19 TE insertions, two correspond to an *Accord* element inserted in the *Cypg6g1* gene that is duplicated in some genomes (Supplementary Data 14). These two insertions are part of an allelic series previously associated with phenotypic variation, in which the more derived the allele is, the greater the level of insecticide resistance[69,70]. We discarded the presence of other structural variants linked to our 18 candidate adaptive TEs that could also be driving positive selection (Table 3 and Supplementary Data 14). Moreover, our set of candidate adaptive TEs was enriched for signatures of selection compared with the whole dataset of TEs present at >5% population frequency (the minimum frequency required to calculate the selection statistics; $X^2$ test, *p*-value = 0.0081). Given the small number of genomes analyzed, strong selection appears to be acting on these 18

insertions as exemplified by the *Accord* insertion[69,70]. However, further functional validation is needed before arriving at any conclusive evidence on the functional role of these TEs. Note that for one of these 18 insertions, we found significant association with the level of expression of the nearby gene in whole-body non-stress conditions (Fig. 6).

We next performed GO enrichment analysis with all the genes located nearby candidate adaptive TE insertions identified so far in *D. melanogaster*, including 84 TEs reported in[33], five other insertions recently described by Bogaerts-Márquez et al.[71], and the 18 TEs identified in this work, including the previously described *Accord* insertion (107 insertions in total). Biological process GO term analysis identified clusters enriched for response to stimulus, behavior, and development and morphogenesis as the ones showing the highest enrichment scores (Fig. 7, Supplementary Data 15). Pigmentation was also among the significant clusters, as has been previously described (Rech et al.[33]). Several gene list enrichments, including regulatory miRNAs and transcription factors, confirmed that genes located nearby these candidate adaptive TEs are enriched for response to stimulus (biotic and abiotic factors), development, behavior, (olfactory and locomotor), and energy metabolism (fatty acid and glucose) functions (Fig. 7 and Supplementary Data 15).

The 107 candidate adaptive TEs identified so far in *D. melanogaster* (Supplementary Data 16a) were enriched for TEs belonging to the *BS* and *Rt1b* families of the LINE order and to the *1360*, *S-element*, *pogo* and *transib2* families of the TIR order (Supplementary Data 16b). Finally, regarding gene body location, we found that the subset of candidate adaptive TEs was slightly enriched for TEs inserted in 5'UTR and promotors, although the differences were not statistically significant (Supplementary Data 16c).

## Discussion

Despite the increasing evidence showing TEs as an important source of genomic structural variation and gene regulation, we are just starting to understand the genome-wide role of these abundant and active components of the genome. The main reasons for this gap in our genomic knowledge are the methodological challenges intrinsic to TEs repetitive nature. New high throughput long-read sequencing technologies that allow to span repetitive regions of the genome, and cutting-edge computational tools offer us now the opportunity to systematically include TE analysis as part of genomics studies. Some works have already demonstrated this, proving that even in an extensively studied biological model organism like *D. melanogaster* we can still identify new and interesting biological properties in which TEs are involved[8,31,32]. In this work, we go a step further by not only using long-read sequencing to generate whole genome assemblies of 32 natural *D. melanogaster* strains collected from 12 populations located in three climate types (Fig. 1 and Supplementary Data 1), but by also taking into account the genetic variability present in these genomes to create a new *D. melanogaster* TE library. We proved that the use of this library —together with a comprehensive TE annotation strategy— not only improves the current gold standard annotation in the well-studied fruit-fly genome (Fig. 2), but also allows the identification of new TE families (Fig. 3) and outperforms state-of-the-art methods for TE annotation using short-reads. Our results also showed that reference genomes consisting of a haplotype-collapse representation are likely to miss some TE insertions as they do not incorporate polymorphisms. Future development of haplotype-resolved de novo assemblies should improve variant calling in long-read genomes[72]. Moreover, the availability of even longer reads together with the improvement of computational analysis

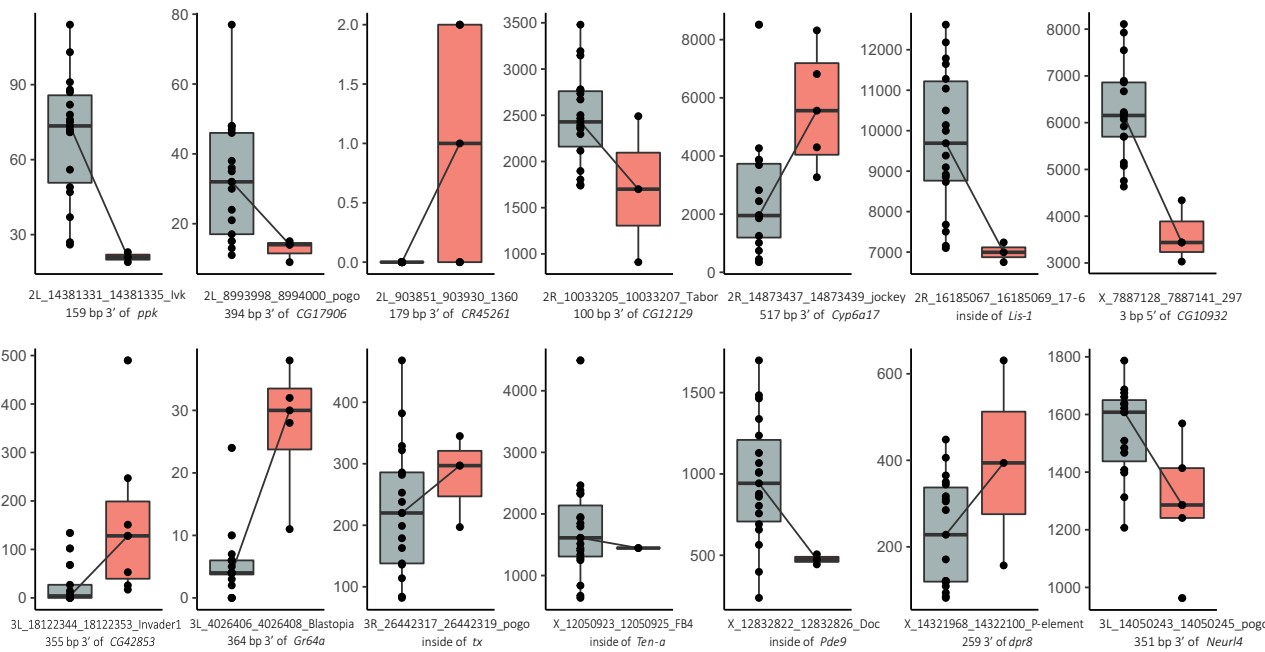

**Fig. 6 Gene expression levels in strains with and without TE insertions.** Gene expression levels in strains without (gray) and with (red) the 13 TE insertions with the most significant association according to our eQTL analysis, and for the *3L_14050243_14050245_pogo* insertion with evidence of selection (last plot). The name of the TE insertions and the genomic location regarding the associated gene is provided. In total, the expression levels of 20 strains are plotted. The boxplot shows median (the horizontal line in the box), 1st and 3rd quartiles (lower and upper bounds of box, respectively), minimum and maximum (lower and upper whiskers, respectively).

**Table 2 TEs showing the highest significance values in their association with the expression of a nearby gene (adjusted *p*-value ≤0.01, defined by an approximation method based on the beta distribution using QTLtools).**

| TE ID | Freq. | Gene symbol | Gene expression | Biological process |
|---|---|---|---|---|
| 2L_903851_903930_1360 | 0.30 | *CR45261* | Up | – |
| 2L_8993998_8994000_pogo | 0.15 | *CG17906* | Down | – |
| 2L_14381331_14381335_lvk | 0.10 | *ppk* | Down | Behavior, Response to stimulus |
| 2R_10033205_10033207_Tabor | 0.10 | *CG12129* | Down | – |
| 2R_14873437_14873439_jockey | 0.20 | *Cyp6a17* | Up | Response to stimulus, Behavior (thermosensory) |
| 2R_16185067_16185069_17-6 | 0.10 | *Lis-1* | Down | Development, Reproduction, Transport/localization, Cell organization/biogenesis, cell cycle/proliferation, Response to stimulus |
| 3L_4026406_4026408_Blastopia | 0.20 | *Gr64a* | Up | Response to stimulus, Nervous system process |
| 3L_18122344_18122353_Invader1 | 0.35 | *CG42853* | Up | – |
| 3R_26442317_26442319_pogo | 0.15 | *tx* | Down | Development, Gene expression |
| X_7887128_7887141_297 | 0.15 | *CG10932* | Down | Small molecule metabolism |
| X_12832822_12832826_Doc | 0.10 | *Pde9* | Down | Response to stimulus, Signaling |
| X_14321968_14322100_P-element | 0.10 | *dpr8* | Up | Nervous system process, Cell organization/biogenesis |
| X_12050923_12050925_FB4 X_12050923_12050925_FB4.t1 | 0.05 0.05 | *Ten-a* | Down | Development, Cell organization/biogenesis, Response to stimulus |

Note that for *Ten-a* gene there were two TEs with equal nominal *p*-value.

should help to characterize nested and highly complex variation in the near future[72].

Improving the annotation of TEs in genome sequences is the first necessary step to accurately evaluate the role of this abundant an active component in genome function and evolution. We identified 472 TEs associated with nearby gene expression variation (Fig. 6 and Table 2 and Supplementary Data 12). While previous genome-wide studies reported an association of TE insertions with reductions of gene expression, our data provide evidence for associations with both up- and down-regulation of nearby genes, in line with a recent analysis on the role of TEs in immune-related genes[73,74]. TE annotations in genomes from arid, temperate and cold climates should allow us to test whether TEs have been involved in adaptation to different environmental conditions. Moreover, the new TE library was also used to annotate 14 other high-quality *D. melanogaster* genomes, which allowed us to analyze the frequency distribution of TE insertions in a total of 47 genomes (Fig. 5). We identified 746 TE insertions present at high population frequencies (≥10% and ≤95%) in genomic regions with recombination rates >0. Eighteen of these

**Table 3 Eighteen candidate adaptive TE insertions showing evidence of selection identified in this work.**

| TE ID | Evidence of selection | Freq | Gene symbol | TE Location | Biological process (experimental evidence) |
|---|---|---|---|---|---|
| 2L_14003409_14003462_Rt1a | nSL | 15% | – | Intergenic | – |
| 2L_8992666_8992668_pogo | nSL | 15% | CG9555 | Intron | NA |
| 2R_11394154_11394156_pogo | nSL | 17% | sprt | Intron | NA |
| 2R_12185376_12185380_accord | nSL | 62% | Cyp6g1 | Promoter | response to insecticide |
| 2R_14078395_14078397_hopper | nSL | 11% | Prosap | Intron | synaptic assembly at neuromuscular junction |
| 2R_18807888_18807894_BS | nSL | 62% | CG15096 | 3UTR | transmembrane transport |
| 3L_12863739_12863742_Transpac | nSL | 19% | CG10943 | Promoter | NA |
| 3L_14050243_14050245_pogo | nSL | 28% | CG6833 | Promoter | NA |
| | | | Neurl4 | Promoter | NA |
| 3L_2426710_2426713_pogo | nSL | 19% | Svil | Intron | NA |
| 3L_3798612_3798621_1360 | nSL | 30% | CG32264 | Intron | NA |
| 3R_20502048_20502058_Doc | nSL | 28% | Dic2 | Promoter | NA |
| | | | CG46441 | Promoter | NA |
| 3R_21385503_21385506_pogo | nSL | 19% | – | Intergenic | – |
| 3R_29952746_29952748_Invader4 | nSL | 23% | TkR99D | Intron | olfactory behavior; detection of chemical stimulus |
| X_15012530_15012533_mdg3 | nSL | 60% | hiw | Intron | autophagy; long-term memory; synapse organization; response to axon injury |
| X_20759991_20759993_BS3 | nSL | 57% | – | Intergenic | – |
| X_2431713_2431716_Doc | nSL | 13% | – | Intergenic | – |
| X_8027468_8027478_Doc6 | nSL | 26% | Tbh | 3UTR | aggressive behavior; behavioral response to ethanol; flight behavior; learning; ovulation |
| 3L_18931204_18931207_F-element | nSL | 15% | CG32204 | Intron | NA |

Biological process information according to FlyBase.

common TE insertions were associated with signatures of selection at the DNA sequence level, including the well-known *Accord* insertion in *Cyp6g1* associated with increased resistance to insecticides, and represent 31% more candidate adaptive TE insertions compared with the previous most extensive analysis[33,69,70] (Table 3). The joint analysis of all the *D. melanogaster* TE insertions showing evidence of positive selection identified so far confirmed that development and response to stimulus are among the most frequent biological processes shaped by TE insertions, together with behavior and pigmentation[33] (Table 3 and Fig. 7).

Overall, given the growing evidence of the importance of TE insertions in genome evolution and function, in addition to their relevance in several human diseases, the approach reported here provides a framework for studying TE dynamics, evolution and the functional implications of TEs in natural population using long-read sequencing. A critical step, was the manual curation of the TE libraries and annotations, a noteworthy effort that allows us to fine-tune the TE annotation strategy to reduce false positives and retain most of the true copies only. We expect that the increasing shift towards the use of long-read sequencing together with comprehensive integration of natural variation in the TE analyses will keep helping to elucidate the role of these active and abundant genome components.

## Methods

**Sequenced strains**. We sequenced the genomes of 32 *D. melanogaster* strains originally collected from natural populations. All the samples represent either isofemale or inbred stocks from such natural populations (Supplementary Data 1). 24 strains were obtained from 11 European natural populations and the remaining eight are RAL strains from the DGRP, obtained from North Carolina, US (Fig. 1, Supplementary Data 1). All flies were reared on standard fly food medium in a 12:12 h light/dark cycle at 25 °C.

**DNA extraction and long-read sequencing**. We sequenced two strains (MUN-016 and TOM-007) using Pacific Biosciences (PacBio) technology and the remaining 30 using Oxford Nanopore Technologies (ONT) and Illumina technologies. DNA for PacBio sequencing was extracted from 400 *D. melanogaster* 5–10 day-old female flies, using the Gentra Puregene Tissue Kit (Qiagen) following manufacturer's instructions. Briefly, 400 flies from each strain were mechanically homogenized in 24 ml of lysis

buffer (proteinase K added) and incubated overnight at 55 °C, and DNA was precipitated with isopropanol after RNAse treatment and protein precipitation. Finally, DNA was resuspended in 1,6 ml of Hydration Solution. DNA concentration was measured using a Nanodrop® spectrophotometer. Most DNA samples for ONT sequencing were extracted from 100 *D. melanogaster* 5–10 day-old female flies from each strain using the Blood and Cell Culture DNA Mini Kit (Qiagen) following manufacturer's instructions with small modifications (Supplementary Data 2; Supplementary Note 1).

PacBio libraries were prepared using 20 Kb SMRTbell and were sequenced using the PacBio RSII System by Macrogen Inc. Korea. ONT libraries were constructed using the Ligation Sequencing Kit (SQK-LSK108 or SQK-LSK109) following manufacturer's instructions (Supplementary Data 2; Supplementary Note 1) and were sequenced *in house* using the MinION device. Basecalling of ONT reads was performed using the *Albacore* Sequencing Pipeline Software (v.2.2). The quality of the long-read sequencing was assessed using *NanoPlot* (v.1.19)[75].

**Short-read sequencing**. The previously extracted DNA used for ONT sequencing was also sequenced using short-read Illumina sequencing either by Macrogen Inc. Korea (TruSeq DNA PCR-free kit, 350 bp insert libraries, 150 bp pair-end sequencing) or by the Genomics Unit of the Center for Genomic Regulation (gDNA-PCR free, HiSeq 2500, 125 bp pair-end) (Supplementary Data 2c).

**Genome assemblies**. We performed de novo genome assembly of the 32 strains sequenced with long-read sequencing technologies. For PacBio sequences, we used *Canu* (v.1.7)[76] for building draft genome assemblies followed by *FinisherSC* (v.2.1)[77] for improving contig continuity. We then aligned PacBio reads to the draft assembly using *pbalign* (SMRT Link v.5.0.1) and used *quiver* (SMRT Link v.5.0.1) to obtain the consensus sequences (polished assembly). PacBio-related programs were all run using default parameters (Supplementary Fig. 12a). For ONT genomes, we also started with *Canu* (v.1.7)[76] with default options for building raw de novo assemblies. We then applied *Racon* (v.1.0)[78], *Nanopolish* (v.0.10.1) (https://github.com/jts/nanopolish) and *Pilon* (v.1.22)[79] for obtaining final polished assemblies (Supplementary Fig. 12b, Supplementary Note 2).

**Genome deduplication, decontamination and scaffolding**. Besides repetitive content, we found that raw de novo genome assembly sizes positively correlated with BUSCO Duplicates (Supplementary Note 3, Supplementary Figs. 13–15). Thus, we evaluated whether levels of heterozygosity might also be involved in determining genome size. Heterozygosity levels in the sequenced strains were evaluated using the short-reads sequences by first calling SNPs against the ISO1 genome following the *GATK* (v.4.0)[80] best practices for variant discovery[81]. Then, we used the *bcftools stats* (v.1.9)[82] for calculating the percentage of heterozygous SNPs at each genome and we found a positive correlation between the estimated heterozygosity and the raw assembly size (Supplementary Note 3, Supplementary Fig. 16). Genomes showing levels of heterozygosity >0.2 were deduplicated

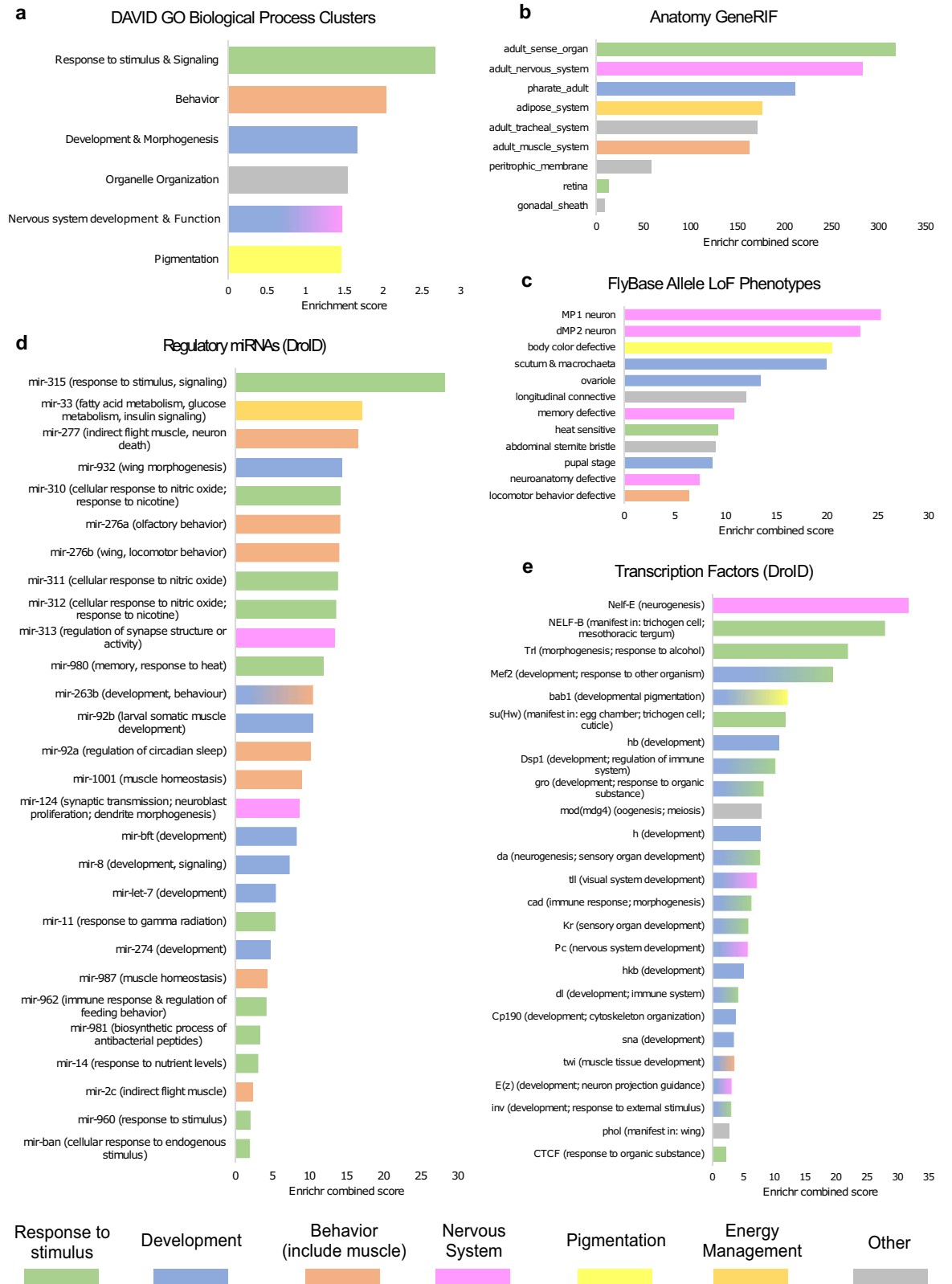

**Fig. 7 Significantly enriched terms for genes nearby 107 TEs showing evidence of selection.** Each panel shows significant enriched terms using different approaches. **a** DAVID GO Biological Process: Horizontal axis represents DAVID enrichment score. Only significant (score > 1.3) and non-redundant clusters are shown. FlyEnrichr results when using different libraries: **b** Anatomy GeneRIF Predicted, **c** Allele LoF Phenotypes from FlyBase, **d** Putative Regulatory miRNAs from DroID and **e** Transcription Factors from DroID. Only statistically significant terms are shown (Fisher test (two sided) adjusted p-value <0.05). Horizontal axis represents the *Enrichr* Combined Score. For Regulatory miRNAs and Transcription Factors, putative biological functions or phenotypes associated were assigned based on FlyBase gene summaries. Bar colors indicate similar biological functions as specified at the bottom of the figure.

(removing alleles -contigs- present twice in the genome) using *purge_haplotigs* (v.1.0.1)[83]; Supplementary Fig. 17, Supplementary Data 3, Supplementary Note 3).

After deduplication, we evaluated contigs for putative contaminations using *MUMmer* (v.4.0)[84]. Briefly, we attempted to align all contigs to the *D. melanogaster* hologenome[85] plus the *D. simulans* genome. We considered as putative contaminant, those contigs showing matches with identities >98% and overlapping >95% of the contig length. We identified putative contaminant contigs in seven genomes (COR-018, LUN-004, MUN-016, MUN-020, RAL-737, TEN-015, TOM-007) (Supplementary Data 3). Once we removed the putative contaminant contigs, we performed a reference-guided scaffolding of the contigs using *RaGOO* (v.1.02)[86], which uses *minimap2* (v.2.9)[87] for aligning contigs to the ISO1 reference genome for ordering and orienting contigs into pseudomolecules. In order to determine whether the scaffolds were covering most of the major chromosomal arms in ISO1, we mapped back the scaffolded genomes to the ISO1 genome using *MUMmer4* (v.4.0)[84]; Supplementary Data 3).

**Assembly quality**. Quality of the assemblies was evaluated by estimating completeness, accuracy and continuity. Completeness and accuracy were calculated using *BUSCO* (v.3.0.2)[88] for the Diptera lineage (diptera_odb9), consisting on 2799 genes. Continuity and completeness were estimated by aligning the polished genome assemblies to the *Drosophila melanogaster* strain ISO1 reference genome release 6[89]. We first masked simple repeats in both genomes using *RepeatMasker* (v.3.0) (www.repeatmasker.org) and then used *MUMmer* (v.3.0)[90] for genome alignment. The quality of the genomes in the context of TEs was evaluated using *CUSCO* (downloaded on May 6, 2020) (Cluster BUSCO; Wierzbicki et al.[36] based on the flanking sequences for 85 out of the 142 annotated piRNA clusters of *D. melanogaster*[91] Supplementary Data 3b, Supplementary Note 5). QV scores were estimated according to Solares et al.[18] using both SNPs and INDELs called from the mapping of Illumina short-reads over the de novo assembled genomes.

**TE sequence accuracy based on long-read sequences**. Incremental updates to the ONT base-calling algorithm has been reported to improve read accuracy[92]. To test whether the ONT base-calling algorithm used in this work affected the TE sequence accuracy, we assembled ONT long-reads available for the reference genome[18] using our pipeline (Supplementary Fig. 12b). We annotated TE copies using the MCTE library and we identified 1842 orthologous TEs comparing with the ISO1 reference genome TE annotation, which represents >83% of the TEs annotated in Solares et al.[18] genome and >89% of the TEs annotated in the ISO1 reference genome. For every TE pair, we performed global pairwise alignments using MAFFT v.7.4 aligner (parameters: *mafft -globalpair -thread 4 -reorder -adjustdirection -auto*). For each pair we then calculated the pairwise identity in two ways: considering and not considering gaps in the alignment. Average gap-ignorant identity was 99.9% and gap-aware identity was 98.9%. Some TE families showed more variability than others but in most cases this variability was explained by individual TE insertions.

**Construction of the Manually Curated TE (MCTE) library**. We used the *REPET* package (v.2.5)[38,40,41] for performing TE annotations using a manually curated TE (MCTE) library of consensus sequences. Briefly, *REPET* is composed of two main pipelines, *TEdenovo* dedicated to de novo detection of TE families and *TEannot* for the annotation and analysis of TEs in genomic sequences. For the creation of the MCTE library, we first run the *TEdenovo* pipeline (default parameters) on 13 genomes (representatives of the geographic distribution of the strains; Table 1). The manual curation of the identified consensuses consisted in three main procedures: removal of redundant sequences, the manual identification of potentially artifactual sequences, and the classification of consensuses into families (Supplementary Note 6). Redundant sequences (consensus sequences present in more than one genome) were removed by first running *PASTEC* (v2.0) with default options[41]. We also performed similarity clustering, multiple sequence alignments (MSA) of the clusters and generated consensus sequences for each MSA in order to obtain a consensus sequence representative of all the genomes (Supplementary Note 6). We manually explored the consensus sequences and their copies using the *plotCoverage* tool from *REPET* and discarded consensuses showing mainly a high number of small copies. The assignation of the consensus sequences into families was performed using *BLAT* (v.35)[93] against the curated canonical sequences of Drosophila TEs from the Berkeley Drosophila Genome Project (BDGP) (v.9.4.1) (https://fruitfly.org/p_disrupt/TE.html). When no matches were found, we used *Repeat-Masker* (v.4)[94] with the release *RepBaseRepeatMaskerEdition-20181026* of the *RepBase*[95] Supplementary Note 6).

**TE annotation**. We use the MCTE library as input for the *TEannot* pipeline to annotate each of the 32 genomes and the ISO1 reference genome. The pipeline was run with default parameters. We annotated TE copies only in the euchromatic regions of the genome since heterochromatic regions are gene-poor[96] and its assembly and annotation usually require specific methods and extensive curation[8,97]. In this work, we determined the euchromatic regions using the recombination rate calculator (RRC)[98] available at http://petrov.stanford.edu/cgi-bin/recombination-rates_updateR5.pl. Such coordinates were originally calculated based on the release 5 of *D. melanogaster* genome so we converted them to release 6

coordinates using the *coord_converter.pl* script from FlyBase[28], resulting in the following regions: 2L:530,000..18,870,000; 2R:5,982,495..24,972,477; 3L:750,000.. 19,026,900; 3R:6,754,278..31,614,278; X:1,325,967..21,338,973. In order to determine the coordinates of the euchromatic regions in each scaffolded genome, we mapped scaffolds to the euchromatic region of the ISO1 genome using *MUMmer* (v3.0)[90]. We then determined the coordinates in the scaffolded genomes by parsing *MUMmer*'s output and extracting the coordinates mapping at the boundaries of the euchromatic region of the ISO1 genome. After running the *TEannot* pipeline over the euchromatic regions of each genome, we performed a post-annotation filtering step consisting in the removal of TE copies <100 bp, as *REPET* cannot accurately annotate these copies, and copies whose length overlapped >80% with satellite annotations.

Multiple sequence alignments of TE insertions for manual curation were performed with *MUSCLE* (v.3.5) using *Geneious* (v.10.0.2) for alignment and visualization (https://www.geneious.com). Identity values between TE copies and the consensus were obtained from *REPET TEannot* pipeline.

**Comparison with short-read-based TE annotations**. We compared *REPET* TE annotations on the de novo assembled genomes using the MCTE library with the annotations performed by two short-read-based TE annotation software: *TEMP* (v.1.05)[45] and *TIDAL* (v.1.0)[46]. To make the comparison unbiased regarding the TE library, we also used the MCTE library for *TEMP* and *TIDAL*. We considered 11 strains representative of the geographic variability and with the best quality assembled genomes (Table 1). We used *BEDtools* (v.2.18)[99] to find the overlapping TE copies predicted by the three different methods (*REPET*, *TIDAL* and *TEMP*) in the 11 strains in a family-aware fashion. To estimate TEMP and TIDAL false negative rate and REPET false positive rate, manual inspection was performed for 300 of the 712 de novo insertions in the COR-014 genome. To do this, we identified the region where each of these TEs was annotated according to REPET/TEMP/ TIDAL and we aligned this region to the ISO1 reference genome to find out if a de novo insertion truly exists. We also used Blast to search for sequence similarities of such genomic region with (i) a database that contains all the individual TE copies identified in our genomes; and (ii) Flybase's 'Transposons - all annotated elements (NT)'. If REPET identified a TE not annotated by TEMP/TIDAL we considered it as TEMP/TIDAL false negative. If a TE was annotated by REPET but we could not find sequence similarities with any of the TE databases by Blast, we considered it as a REPET false positive. Additional 50 TEs annotated by TEMP/ TIDAL but not by REPET were also manually curated following the same procedure.

**TE orthology identification**. To identify orthologous TEs, we first transferred the TE coordinates from each strain to the ISO1 reference genome. Briefly, we used a similarity and synteny approach based on *minimap2* (v.2.9)[87] mapping of the TE sequence and its flanking regions to the ISO1 genome and the coordinates of genes as anchored synteny sequences (see Supplementary Note 11 for details). To transfer the TEs, we took into account whether its flanking region mapped unequivocally or not, whether it mapped completely or partially, whether it was a tandem or nested TE, among others. Then, based on the information of the alignment and characteristics of the transfer, we defined each of the TEs as either reliable or unreliable, being the latter ones discarded from the transfer. Finally, once all the reliable TEs of each strain were transferred to the reference, the orthologous TEs were defined (Supplementary Note 11, Supplementary Figs. 18–20). To avoid false positives, we only used those TEs for which more than half of the orthologous TEs were larger than 120 bp. All scripts used for the TE transfer are available at www.github.com/sradiouy/deNovoTEsDmel.

After determining the presence/absence of TEs, we classified them in three frequency classes: rare (TEs present in <10% genomes), fixed (TEs present in >95% of the genomes) and common (TEs present in ≥10% and ≤95%). We then calculated the number of TEs for each frequency class considering different number of genomes, starting from 5 up to 47. We estimated the mean and standard deviation of the number of TEs in each frequency class by randomly choosing genomes (30 iterations). Then, we intersected the different sets of common TEs considering 10, 20, 30, 40 and 47 strains using *UpSetR* (v.1.3)[100] and also established different sets of TEs based on the geographical origin of the genomes and compare them using *VennDiagram* (v.1.6)[101]. For determining the location of the TE insertion regarding annotated genes, we used *annotatr* (R package version 1.20.0).

**TE eQTL analysis**. In order to identify polymorphic TEs significantly associated with the expression levels of nearby genes, we analyzed available whole-body RNA-Seq data from 12 European[56,57] and 8 American strains[55] (Table 1, Supplementary Data 2c). Briefly, RNA-Seq data was trimmed using the *fastp* package (v.0.20)[102] with default parameters. Expression levels were quantified by applying the *salmon* package (v.1.0.0)[103] against the ENSEMBL (Dm.BDGP6.22.9) transcripts. Obtained transcripts per million (TPM) were summed up to gene level and *rlog* normalized using *DESeq2* (v.1.28.1)[104]. eQTL analysis was performed using the *QTLtools* package (v.1.2)[105] taking into account the population structure (Supplementary Figs. 21 and 22). Putative cis-eQTL were searched within a 1 Kb window around each gene using the *cis* module in QTLtools. We used the nominal pass to evaluate

the significance of the association of the gene expression level to TE insertions. The genotype table was created with a custom script. Finally, we performed a permutation pass (100,000 permutation) to adjust for multiple testing. Overall, we evaluated 12,281 eGenes-TE involving 4709 genes and 9676 TEs. We focused on TEs located in high recombination regions and we considered significant eGenes-TE associations when the nominal $p$-value and the associated adjusted $p$-value were significant (<0.05). Manual inspection of the 15 TEs that were the top variant and the most significant associations (adjusted $p$-value <0.01) confirmed that they were correctly annotated in all the genomes (300/300 correct calls) except for an INE-1 element that was removed from the analysis as it was fixed in all the genomes analyzed (7/20 correct calls) and a Blastopia insertion that was miss annotated in one of the strains (19/20 correct calls).

**Positive selection analysis.** We looked for evidences of selection in genomic regions targeted by TE insertions using *selscan* (v.1.2.0a)[106] and Single Nucleotide Polymorphisms (SNPs) as a proxy (Supplementary Note 12). We looked for evidences of incomplete soft or hard selective sweeps in the 46 *D. melanogaster* genomes (the 32 sequenced in this work plus the 14 genomes sequenced by Chakraborty et al.[8]). SNPs were called using the *GATK* (v.4.0)[80] *HaplotypeCaller* best practices for variant discovery[81] and the haplotype phasing was performed using *SHAPEIT4* (v.4.1)[107]. Initial SNP calling resulted in 5,578,437 SNPs, from which we kept only biallelic SNPs using the GATK command *SelectVariants* (parameters *-select-type SNP -restrict-alleles-to BIALLELIC*). Finally, we also removed SNPs with *missing data* in at least one genome, resulting in a total of 2,797,589 SNPs (available at https://doi.org/10.20350/digitalCSIC/13708). Genetic positions and recombination maps[108] were obtained from FlyBase (https://wiki.flybase.org/wiki/FlyBase:Maps, last updated June 15, 2016). Three statistics were calculated in *selscan*: iHS[65], iHH12[66,67] and nSL[68]. iHS and nSL statistics are both aimed to identify incomplete sweeps, where the selected allele is not fixed in the sample, and the main difference is that nSL is more robust to recombination rate variations, which increases the power to detect soft sweeps. iHH12 has been developed for the detection of both hard and soft sweeps, with more power than iHS to detect soft sweeps[106]. After obtaining results from each statistic, we normalized them using the *norm* package in 10 frequency bins across each chromosome. We considered iHS, iHH12 and nSL normalized values to be statistically significant for a given SNP if they were greater than the 95th percentile of the distribution of normalized values for SNPs falling within the first 8–30 base pairs of small introns (≤65 bp) which are considered to be neutrally evolving[109] Supplementary Data 13b). In order to identify TEs putatively linked to the selective sweeps, we analyzed the co-occurrence (in the same strains) of the allele showing signatures of a selective sweep and a nearby TE (<1 Kb). We focused only on those TEs more likely to have a role in adaptation: First, from the 28,365 transferred TEs, we selected those at frequencies ≥10% and ≤95% and inserted in regions with recombination rates >0, as these insertions are more likely to play a role in adaptive evolution rather than being linked to the causal mutation[33], resulting in 902 TEs. From those, we also discarded TEs belonging to the INE-1 and the LARD families, since those represent very old TE families likely to have reach high frequencies neutrally, ending up with a set of 746 TEs. We considered TEs in this 746 dataset as likely to be enriched for candidate adaptive TEs[33]. We then looked whether any of these 746 TEs was nearby a SNP showing significant values at some of the haplotype-based selection test. Finally, for each SNP-TE pair we established criteria of 'co-occurrence' by requesting certain number of the strains containing both the SNP allele undergoing a selective sweep and the nearby TE: for TEs present in 5-6 strains we request at least 4 of the strains to contain both the allele undergoing a selective sweep and the nearby TE and for TEs present in ≥7 strains we request the majority of strains to contain both the significant SNP and the nearby TE. In all cases, we also requested the TE to be absent in 100% of strains that do not contain the significant SNP allele (Supplementary Data 13a).

To discard that other CNVs could be linked to the identified 18 TEs associated with signatures of selection, we identified using the *Structural Variants and MUmmer* (SVMU) tool the presence of CNVs in the 1 kb regions flanking these insertions (Supplementary Data 14)[8].

**TE genomic location.** TE's overlapping genes or located nearby genes were determined using the following criteria: (i) we considered only protein-coding genes from FlyBase gene annotation r6.31 (13,939 genes); (ii) to determine the gene location (3′UTR, 5′UTR, CDS, INTRON, PROMOTER) we considered the position regarding the longest transcript only; (iii) promoter regions were considered as the 1 Kb region upstream of the TSS; (iv) 3′UTR, 5′UTR, CDS, INTRON coordinates were obtained from the header of the *fasta* files available at FlyBase (http://ftp.flybase.net/genomes/Drosophila_melanogaster/dmel_r6.31_FB2019_06/fasta/); (v) only the closest gene (<1 Kb) to the TE was considered; (vi) when a TE overlapped (distance = 0) with more than one gene, all overlapping genes were considered. This is also true for the (rare) case in which the distance to more than one gene is exactly the same; and (vii) when no gene was found at <1 Kb, the TE was classified as 'Intergenic'.

**Enrichment analysis.** GO enrichment analyses for list of genes nearby candidate TEs were performed using DAVID functional annotation cluster tool (v.6.8)[110,111]

using all *D. melanogaster* protein-coding genes from FlyBase gene annotation *r6.31* as a background. In addition, we also used the online version of FlyEnrichr[112,113] to analyze enrichments regarding four gene-set libraries: 1) *Anatomy GeneRIF Predicted*: list of genes with predicted GeneRIF terms involved in fly's bodily structures (Gene Reference into Function: https://www.ncbi.nlm.nih.gov/gene/about-generif). 2) *Allele LoF Phenotypes from FlyBase*: FlyBase's allele phenotypic dataset. Loss of function phenotypes and gene sets with alleles producing those phenotypes. 3) *Putative Regulatory miRNAs from DroID*: DroID's (http://www.droidb.org/) putative miRNA targets dataset and 4) *Transcription Factors from DroID*: DroID's (http://www.droidb.org/) transcription factor-gene interactions datasets. We report only terms with an adjusted $p$-value <0.05.

**Reporting summary.** Further information on research design is available in the Nature Research Reporting Summary linked to this article.

## Data availability

All scaffolded assemblies and the raw data (long and short read sequencing) have been deposited in NCBI database under the BioProject accession PRJNA559813. The VCF file containing SNP callings for 46 *D. melanogaster* genomes used for testing positive selection evidences is available at DIGITAL.CSIC repository (https://doi.org/10.20350/digitalCSIC/13708) Fasta sequences for the *D. melanogaster* Manually Curated Transposable Elements (MCTE) library are available at DIGITAL.CSIC repository (https://doi.org/10.20350/digitalCSIC/13765). The new consensus sequences are deposited in Dfam (Storer et al.[114]). Recombination rates according to Fiston-Lavier et al.[98] and Comeron et al.[108] for *D. melanogaster* genome release 6 are available at DIGITAL.CSIC repository (https://doi.org/10.20350/digitalCSIC/13766). BED files containing Transposable Element (TE) annotations for 47 Drosophila melanogaster genomes are available at DIGITAL.CSIC repository (https://doi.org/10.20350/digitalCSIC/13894).

## Code availability

All scripts and codes have been deposited to GitHub and are freely accessible from https://github.com/gabyrech/deNovoTEsDmel and https://github.com/sradiouy/deNovoTEsDmel.

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

## Acknowledgements

We would like to thank DrosEU researchers for sharing with us the strains sequenced in this manuscript (Supplementary Data 1). We also thank Miriam Merenciano, Anna Ullastres, and Lain Guio for helping create the inbred strains. This project has received funding from the European Research Council (ERC) under the European Union's Horizon 2020 research and innovation programme (H2020-ERC-2014-CoG-647900). S.R. was funded by the MICINN/FSE/AEI (PRE2018-084755) and VH was funded by the Generalitat de Catalunya (FI2017_B00468). DrosEU is funded by an ESEB Special Topic Network award.

## Author contributions

G.E.R.: design of the work, data acquisition, analysis and data interpretation, drafted and revised the manuscript. S.R. and S.G.-R.: data acquisition, analysis and data interpretation, drafted and revised the manuscript. L.A., V.H., L.G and H.L.: data acquisition and revised the manuscript. V.J. and H.Q.: data analysis and revised the manuscript. J.G.: conception and design of the work, analysis and interpretation of data, drafted and revised the manuscript.

## Competing interests

The authors declare no competing interests.
