## [Peer Review File · Nature Communications]

Population-scale long-read sequencing uncovers transposable elements associated with gene expression variation and adaptive signatures in *Drosophila*Reviewers' Comments:

Reviewer #1:

Remarks to the Author:

In this manuscript, Rech and colleagues investigate the transposable element (TE) landscape in the genome of the model organism *Drosophila melanogaster*. The significant advance here, over previous studies, is the use of multiple long-read genomes sampled from multiple populations throughout the world, to identify and construct a TE library, rather than the reliance on a single high-quality (i.e., the reference) genome. Additionally, the authors are able to relate population variation in TEs to population variation in gene expression and signals of adaptive evolution.

Overall, I think the authors have done an impressive job identifying, characterizing, and curating TEs in these *D. melanogaster* genomes, and show that even in one of the best-studied model organisms, many things can be discovered with the application of new technologies. However, I have some concerns about quality assessment of these long-read genomes and crucially the claim that many TEs are implicated in adaptive evolution. Additional clarifications in the manuscript or some updated analyses would alleviate these concerns. Lastly, I have a stylistic suggestion for how the manuscript might be improved: specifically, I found the discussion to be under-developed in that it mostly reiterates the results of the manuscript rather than providing interpretation or insight. I will explain my concerns in detail below.

Major concerns:

1. Quality assessment of long-read assemblies

A well-known concern with Oxford Nanopore (ONT) long-read sequencing is the noisy error profile of ONT long reads, particularly indels in homopolymer runs [1,2]. It also seems that the authors used an older, now deprecated version of ONT's base calling software which if I am not mistaken has a much higher error rate than the more recent base callers. While sequences that have been well-polished with both long and short reads should be relatively accurate [2], repetitive elements like TEs are notoriously difficult to polish since shorter reads will map ambiguously to these regions [3].

Perhaps this does not matter so much for building consensus sequence and orthology inference, since the sequence accuracy of an individual TE does not matter so much. However, the authors do not really try to quantify the accuracy of TE sequences, and this seems like it would be limiting to any future work that tries to utilize the resource that the authors have put so much effort and care into. I can appreciate how difficult it is to do this in the absence of a "truth" sequence for the wild-caught strains, but the *D. melanogaster* reference genome is well-characterized and Nanopore data of the same accuracy as the authors' exists for the reference strain [4] (note: if I'm not mistaken, there is no difference in the accuracy of ONT data called with Albacore 2.0.2 and Albacore 2.2).

Perhaps the authors could re-assemble the data of [4] with their pipeline and then, after aligning their de novo assembly to the reference genome, characterize the sequence accuracy in TEs. I also thought it would be helpful for the authors to include comparisons to the reference genome as a benchmark for example in Table 1.

On a related note, I found the scaffold CUSCO [5] scores (p.5 lines 122-128) to be a questionable metric for quantifying the quality of TE sequences. To me, the CUSCO metric makes sense when computed for contigs: then, it is an assessment of how well some particularly challenging regions were assembled. However, the authors performed reference-based scaffolding for all their assemblies and reported a scaffold CUSCO (sc.CUSCO) score. In this context, the sc.CUSCO score is more of a measurement of how well the unique sequences flanking piRNA clusters are assembled and if they map to the reference, but also depend on how well piRNA clusters are assembled in the reference. The challenging sequences in the piRNA cluster do not need to be contiguously assembled to obtain a good sc.CUSCO score. I would speculate that if the flanking sequences are sufficiently complex and unique

enough to be assembled with just short reads, a reference-scaffolded short-read assembly would also have a very high sc.CUSCO score.

2. Performance of long-read versus short-read methods (specifically, page 9 lines 237-299).

I thought this section was very confusing to read. At first glance, the header suggests that annotations based on long-read genomes (REPET) significantly outperform short-read methods (TIDAL, TEMP) [6,7] by 60%, but throughout the section, the numbers seem quite similar and it is suggested that the haploid genome used for the REPET approach is also limiting. If you take the numbers presented in the last paragraph then you get ~60%, but that's conditioning only on the 92 TE insertions uniquely detected by REPET out of the thousands that were detected.

Another thing that was unclear was how the authors performed the comparison against TIDAL and TEMP [6,7]. Were the databases for those two programs the original databases from the TIDAL and TEMP studies? If the REPET library was built with the same genomes that TE identification was performed in, but the TIDAL and TEMP libraries were built from different samples, wouldn't REPET naturally perform better than TIDAL or TEMP in this scenario?

This does not seem to strongly support the claim that the authors' approach "outperforms state-of-the-art methods for TE annotation using short-reads" (page 16, lines 483-484). I think that significant clarification to the methods and/or approaches to make this a more similar comparison (i.e. ensuring the same strains were used to build TE libraries across all methods) is necessary to demonstrate that this is so.

I also did not like that the terminology for the frequency classes (unique, polymorphic, fixed) was not defined before its usage nor in the Figure 4 caption.

3. Characterization of TE population variability (specifically, page 12 lines 346-394)

I thought the way population variability in TEs is characterized could be better connected to the demographic history of *D. melanogaster*. Currently, this manuscript characterizes the counts of TEs in certain frequency classes (Figure 6) and gives some clues into the geographic distribution of TEs but does not provide any additional insight into these patterns.

One thing that immediately comes to mind is how the patterns of TE variability are similar or different to the patterns of SNPs across *D. melanogaster*. The consistent linear increase in the number of rare annotated TEs in Figure 6A seems to be consistent with the star-shaped genealogy of a recently expanded population, but does purifying selection on TEs also contribute to this? How does the frequency spectrum of TEs compare to nonsynonymous SNPs, synonymous SNPs, SNPs in conserved regions, "neutral" SNPs, etc.?

I will acknowledge that this is a big ask and these sorts of analyses may not be straightforward and outside of the scope of a paper revision/more appropriate for a new paper. However, the authors have emphasized the importance of understanding TE variability in natural populations in this manuscript. Providing at least a simple link between TE diversity and the population genetic forces in *D. melanogaster* (typically studied with SNPs) would help make this point.

4. The role of TEs in adaptation (specifically p14-15 lines 429-462)

I don't know if the authors have convincingly demonstrated that these TEs are candidates for adaptation.

The authors took a set of 479 TEs at >10% frequency, performed selection scans in flanking regions, and found that 34 of these TEs were associated with a signal of positive selection. The methodology is

a bit unclear (p.23-24, lines 685-709). Was the calculation of the statistics only performed with haplotypes with TEs present as lines 708-709 seems to imply? If not properly stratified across frequency classes, wouldn't the variance of these statistics differ across different TE frequencies, meaning the tails might be enriched for data points from a low frequency TE/high variance distribution?

Either way, a null expectation for the association between TEs and signatures of adaptation is not provided, so it is difficult to distinguish whether this is simply an overlap between two random, independent processes or whether a correlation (or a causal link) exists. *D. melanogaster* is notorious as a species in which adaptation occurs frequently. If one were to randomly choose 479 SNPs/loci in the *D. melanogaster* genome (ideally, matching for background characteristics like allele frequency, recombination rate, or the density of functional elements) and perform a similar scan, what proportion of these loci would you expect to see a signature of adaptation associated with? What would the enrichment in GO terms look like for the null sets?

Given that the role of TEs in adaptation is a central claim of this paper (it's in the title), a stronger link between TE insertions and adaptation needs to be established.

5. The discussion is underdeveloped

I thought this section could be better developed. As it stands, the Discussion consists of brief summaries of the work (e.g., better TE annotations) or broad, sweeping statements (e.g., long reads are useful, studying TEs may provide new insights). I wish the authors would discuss some specific implications of their results, how future studies possibly with even longer reads might open up new avenues for analyses, how analyzing data across different climate zones or across different species using their approach would be insightful, etc.

Minor comments:

1. The authors repeatedly state that their samples are from "five climatic regions" throughout the manuscript, but do not provide any rationale for why this kind of sampling is important, or how climatic regions relate to any of their results.
2. Page 5 lines 118-119: Did differences in read length explain differences in genome size or TE content?
3. Figure 6 caption: please explain what Figure 6B shows rather than just saying it's an "UpSetR plot."
4. Figure 7 caption: please fully write out what each library is rather than using notation like "Allele_Phenotypes_from_FlyBase_2017"
5. There are a number of minor typos throughout the manuscript.

References:

1. Watson et al. 2019, <https://www.nature.com/articles/s41587-018-0004-z>
2. Koren et al. 2019, <https://www.nature.com/articles/s41587-018-0005-y>
3. https://nanoporetech.github.io/medaka/draft_origin.html
4. Solares et al. 2018, <https://academic.oup.com/g3journal/article/8/10/3143/6026978>
5. Wierzbicki et al. 2020, <https://www.biorxiv.org/content/10.1101/2020.03.27.011312v1.full>
6. Zhuang et al. 2014, <https://academic.oup.com/nar/article/42/11/6826/1431591>
7. Rahman et al. 2015, <https://academic.oup.com/nar/article/43/22/10655/1803983>

Reviewer #2:

Remarks to the Author:

This is an important paper that uses long read sequencing (PacBio and Oxford Nanopore) to assemble 32 *D. Mel* genomes and semi-manually (with the help of automatic pipelines) curated a library of transposable elements (TEs). This library is then annotated against all 32 + 15 (from published study) genomes generated with long read sequencing to identify fixed and polymorphic TEs. For those with RNA-Seq data, the polymorphic TEs are tested for association with expression of genes nearby (eQTL analysis). Finally, TEs are assessed for signature of positive selection. Overall, the paper is very well written, addresses a significant gap in our understanding of genome structure variation in *Drosophila*, and produces a useful resource (the TE library, especially the novel ones).

I have a few comments, mainly regarding data presentation, to improve readability of the paper:

1) The saturation analysis (what I call Figure 6) could be improved or done differently. Apparently the classification (frequency based) is variable depending on what strains are used to identify the TEs. I would fix the classification based on the full data set, then perform the same analysis again to draw Figure 6.

2) I'm worried about the fact that nearly all TEs associated with expression were from the new TEs identified in this study. The eQTL analysis of the paper lacks some details. I would suggest pick a few TEs (significant and randomly selected non-significant ones):

- a) draw gene structure (with exons) along with the insertion sites of the detected TEs
- b) superimpose wiggle grams on them
- c) also draw a scatter plot of the estimated expression versus TE presence/absence

I commend the authors for being very careful in the TE annotation work, but this part lacks a bit of rigor.

3) For the gene set enrichment analysis, a more detailed description is needed in the Methods. For example, what is the background gene set? What is considered significant? What kind of test is used?

4) Title could be better. First, the title should reflect TE rather than SV. Second, the relationship with adaptive evolution is weak.

Minor comments:

- Figure 6. Why as the number of randomly sampled strains increases (from top to bottom, 10->47), the number of identified TEs decreases? Shouldn't it be the opposite trend?
- Line 98: please in this section note that 2 strains were sequenced using PacBio and 30 by MinION. This is an important piece of information that should be in the main text too.
- Line 540: accessed -> assessed
- Figure S3 title: mayor -> major, also I would call these dot plots rather than mummer plots.
- On my reviewer page, I cannot find any of the supplemental files. These should somehow be made available.

Reviewer #3:

Remarks to the Author:

Repetitive sequences in the genome play an important role in genome and phenotypic evolution, yet much of them are inaccessible to the short reads. This limitation has impeded systematic functional and evolutionary analysis of the repetitive genomic sequences. In "Hidden intraspecies structural variation contributes to adaptive evolution in *Drosophila melanogaster*" Rech et al. report de novo

genome assembly of 32 *D. melanogaster* strains from Europe and North America. They assembled genomes using Oxford Nanopore long reads and then created a custom repeat library to comprehensively map the transposable elements in the assembled genomes. Their analysis yielded 3 new TEs in the well-studied model organism *D. melanogaster*. They further showed that short reads based methods of TE annotation fail to detect 60% of the TEs they found in the assemblies. Among the TEs they detected in their assemblies, several are located in genomic regions showing signatures of positive selection. Finally, they show that >400 TEs are associated with gene expression variation, suggesting that TEs are a potential source of gene regulatory variation.

The 32 new de novo genome assemblies reported by the paper will be a useful resource for the *Drosophila* community and population geneticists in general. The result on novel TEs is interesting and so are the newly uncovered putatively adaptive TEs. However, many details and citations of prior work in the relevant sections are missing, making it harder to evaluate the claims by the authors about the discovery of new TEs and their adaptive and functional significance. The paper would benefit greatly from providing these details and clarifications. My detailed comments are below.

The criteria for calling a TE hidden vs visible was unclear. The authors need to clearly mention the criteria used for hidden vs visible/novel TEs in all contexts they are mentioned in the paper (e.g. short reads vs assembly comparison, putatively adaptive TEs, putative cis-regulatory TEs). Are TEs being described as variants or just genomic elements? In some sections, authors describe the TEs as genomic elements (TE annotation) but in other sections they are treated as variants (e.g. putative adaptive variants). Were the new TE families identified by the authors present in the 14 reference *D. melanogaster* genomes previously sequenced (doi: 10.1038/s41467-019-12884-1)? Were the variants hidden because they escaped detection by TE genotyping pipelines based on paired-end short reads (e.g. PMID 23883524)? Due to this lack of clarity, it was not immediately clear whether novel TEs described in the sections on signature of adaptation and gene expression are novel because they could not be detected with short reads or because these genomes were not studied before. At the very least, the authors should support their claims about novel TE variants by comparing their calls to a short reads based genotyping pipeline (e.g. PMID 23883524) and show that those TEs were invisible to the short reads.

The method of detecting candidate adaptive TEs was a bit unclear. The Methods section says that the authors used SNPs for the tests of selection, but in the main text they present the TEs as the putative adaptive variants. How did the authors narrow down their search from a collection of SNPs to a particular TE? Were the TEs encoded as variants in their analysis and TE variants showed zero LD with the adjacent SNPs? This needs to be clarified because this is one of the major results of their paper. I am particularly curious about how the authors disentangled the candidate TEs from linked SNPs or other small or large SVs at the focal regions showing the signatures of positive selection. The claim of potential adaptive significance of the TEs would be strengthened if the authors provided the gene expression phenotypes of their candidate adaptive TEs.

The authors missed citation of prior work in several sections. Citing prior work would not only acknowledge the work people have already done, but will also place the results described in the manuscript in a larger context, strengthening the claims of novelty. One prominent omission was the citation of the work Julie Cridland did on showing the association between TEs and gene expression in DGRP (doi: 10.1534/genetics.114.170837). Did the authors sequence any of the lines Cridland et al. used in their study? If they did, it would be interesting to know whether any of the variants that Cridland et al. missed in the DGRP strains was detected by the new assemblies and showed association with gene expression variation. Similarly, the extent to which TE insertions are missed by short reads in *D. melanogaster* was first quantified and reported by Chakraborty et al. (doi: 10.1038/s41588-017-0010-y). On a related note, a high frequency copy number variant (deletion) of *Cyp6a17*, showing a large effect on temperature preference behavior, has been reported previously (doi: 10.1038/s41588-017-0010-y). I am curious about the status and role of this CNV in the gene expression variation the authors observe for *Cyp6a17*. Finally, I was wondering about why the authors

do not cite any paper from *D. melanogaster* that has been published in the last five years when they discuss the limitations of short reads in discovery of genetic variation due to repetitive sequences in the first paragraph of Introduction.

The site frequency spectrum of TEs suggest that on average TE insertions are strongly deleterious in *D. melanogaster* (doi: 10.1038/s41467-019-12884-1). It is therefore expected that most polymorphic TEs are rare and the vast majority of the TE variants will not be shared by individuals. Therefore, the motivation for the section on most common variants can be sampled by 20 individuals is not clear. In this section, I think the authors meant to say 'only additional 36' instead of "only 36" at Line 374. Same goes for "27 TEs" at line 375. I thought the most interesting part in this section was the TEs showing evidence for their roles in local adaptation, but relatively less real estate is spent on this result.

Please provide the error rate of the assemblies at the nucleotide level (QV). Details on how to estimate QVs can be found in Solares et al. (doi: 10.1534/g3.118.200162).

The title of the paper is misleading. The authors do not show conclusively that TEs are the causal adaptive variants, making the claim about newly discovered TEs contributing to adaptation only speculative. Additionally, the authors discuss only one class of SVs (TE insertions) in the paper, but the title gives the impression that the paper is about all classes of SVs.

It was surprising that reference-guided scaffolding drastically improved the BUSCO scores of the assemblies (64.1% to 93.7%). Surprising because scaffolding does not really add any new sequences to bridge gaps (or resolve repeats) in the assemblies. Could the authors comment on why scaffolding improved the BUSCO scores so drastically?

Does RepeatMasker find any hit based on the latest release of the *Drosophila* TE library at the genomic regions where the newly discovered TIR element is found? Or, does it report nothing in those regions?

The status of heterozygosity in the strains used in the expression GWAS experiment is not clearly stated. Were these strains inbred? If they are not inbred, did the authors take the heterozygosity of the strains into account while doing the association study? A related question is: how did the authors perform haplotype based selection tests in heterozygous strains, where the number of segregating chromosomes are unknown?

Reviewer #4:

Remarks to the Author:

Illumina and other short read technologies are not ideal for inferences about Transposable Elements, and for this reason this group decided to apply long-read sequencing methods on a set of 32 *D. melanogaster* strains. The project is well motivated and results in 32 high quality genome sequences of *Drosophila melanogaster* from 12 geographical populations. The authors applied Oxford Nanopore technology for 30 of the genomes, and PacBio for 2 of them. The procedures all appear to be state of the art, and the assembly procedures all reasonable. In the end there was a huge range of N50 statistics, going from 400 kb to 19 Mb. The authors claim that this seemed to make little difference to BUSCO scores or TE call accuracy, but the latter seems a bit anecdotal. Much of the analysis centers of inference of Transposable Element inference, an aspect of genome assemblies where short reads are particularly poor, and this problem is largely resolved by both long read technologies. The paper provides a solid technical presentation of the genome assemblies and inferences of TE insertions, including inference of effects on altered gene expression and detection of positive selection. A few specific technical issues follows:

1. Regarding the calling of TEs, it is probably worth running RepeatModeler2 as well. Claims in Flynn et al. (PNAS 2020) make it appear to perform very well.
2. Regarding the claim that the long read assemblies detect 58% more insertions, this reader wonders whether it would be useful to provide more definitive validation of these insertions, such as PCR. This does not seem to be the standard in this area, so it seems that confidence in the assemblies is very high, but surely at some point direct validation was done.
3. In the section on inference of TE insertions associated with changes in gene expression, this reader is not sure whether the standard method of PCA covariates is adequate to control for population structure, especially when the population structure is as great as it is with *D. melanogaster*, and such small samples from so many populations is included. Some added justification seems warranted.
4. Lines 237-299 A LOT of space devoted to comparison of calls of REPET, TIDAL and TEMP. This seems like material that could better be relegated to the Supplement. The main point that short reads miss 60% of TEs should be in main text.
5. Lines 301-344 – The section discussing the similar numbers of TEs per line is rather qualitative, and should include well known population genetic metrics for heterogeneity among lines and among populations. Is there any theoretical expectation for these numbers, given ballpark estimates of rates of transposition.
6. Line 346-394: The focus in this section is on subsampling to see what is the sample size needed to capture most common TEs. The claim that 20 is sufficient again seems to lack quantitative rigor. Similarly, the count of population-specific TEs depends on their frequency spectrum and the sample sizes.
7. Lines 396-427 center on functional effects associated with TE insertions, highlighting a role for TEs as cis-eQTLs. This comes across as mostly being anecdotes and not much new. Given this more exhaustive sample than before, can we make conclusions about the fraction of TE insertions expected to impact gene expression, whether different classes of TEs have different effects, and whether there is any signature of selection as a result of these effects?
8. In Lines 429-462 the authors report that some 36 TEs in the set display evidence for positive selection, as assessed by tests identifying flanking haplotypes that appear to have expanded in frequency (e.g. the iHS test). These are awfully small sample sizes for inference of selection, so perhaps a word is warranted about the power of these tests, and the resulting conclusion that rather strong selection appears to be acting on these positive cases.

A couple of trivial edits:

Line 469 change "methodical" to "methodological"

Line 692 change "mayor" to "major"

Reviewers' comments:

Reviewer #1 (Remarks to the Author):

In this manuscript, Rech and colleagues investigate the transposable element (TE) landscape in the genome of the model organism *Drosophila melanogaster*. The significant advance here, over previous studies, is the use of multiple long-read genomes sampled from multiple populations throughout the world, to identify and construct a TE library, rather than the reliance on a single high-quality (i.e., the reference) genome. Additionally, the authors are able to relate population variation in TEs to population variation in gene expression and signals of adaptive evolution.

Overall, I think the authors have done an impressive job identifying, characterizing, and curating TEs in these *D. melanogaster* genomes, and show that even in one of the best-studied model organisms, many things can be discovered with the application of new technologies. However, I have some concerns about quality assessment of these long-read genomes and crucially the claim that many TEs are implicated in adaptive evolution. Additional clarifications in the manuscript or some updated analyses would alleviate these concerns. Lastly, I have a stylistic suggestion for how the manuscript might be improved: specifically, I found the discussion to be under-developed in that it mostly reiterates the results of the manuscript rather than providing interpretation or insight. I will explain my concerns in detail below.

Major concerns:

1. Quality assessment of long-read assemblies

A well-known concern with Oxford Nanopore (ONT) long-read sequencing is the noisy error profile of ONT long reads, particularly indels in homopolymer runs [1,2]. It also seems that the authors used an older, now deprecated version of ONT's base calling software which if I am not mistaken has a much higher error rate than the more recent base callers. While sequences that have been well-polished with both long and short reads should be relatively accurate [2], repetitive elements like TEs are notoriously difficult to polish since shorter reads will map ambiguously to these regions [3].

Perhaps this does not matter so much for building consensus sequence and orthology inference, since the sequence accuracy of an individual TE does not matter so much. However, the authors do not really try to quantify the accuracy of TE sequences, and this seems like it would be limiting to any future work that tries to utilize the resource that the authors have put so much effort and care into. I can appreciate how difficult it is to do this in the absence of a "truth" sequence for the wild-caught strains, but the *D. melanogaster* reference genome is well-characterized and Nanopore data of the same accuracy as the authors' exists for the reference strain [4] (note: if I'm not mistaken, there is no difference in the accuracy of ONT data called with Albacore 2.0.2 and Albacore 2.2).

|

|

Perhaps the authors could re-assemble the data of [4] with their pipeline and then, after aligning their de novo assembly to the reference genome, characterize the sequence accuracy in TEs. I also thought it would be helpful for the authors to include comparisons to the reference genome as a benchmark for example in Table 1.

>>>We thank the reviewer for their comments. We used the latest caller available at the time we were doing the base calling and we used short reads, as the reviewer mentions, to polish the sequences. The reviewer is correct that new base callers have been released since we performed our analysis. The most recent comparative analysis of ONT's base calling software that we could find was published in June 2019 by Wick et al (<https://genomebiology.biomedcentral.com/articles/10.1186/s13059-019-1727-y>). The consensus identity and read identity with Albacore v2.2. (the one we used in our work) are similar or even higher than that of some even more recent basecallers (Figure 1 of Wick et al 2019). We are thus confident that the results presented in our manuscript are high-quality.

Sequence accuracy of TEs based on long-read sequencing has been assessed in Wierzbicki et al 2020. While it is reasonable to expect that polishing of the long-reads could eliminate polymorphisms from TEs sequences, the authors showed that this is not the case. Note that Wierzbicki et al (2020) performed the base calling using Albacore v2.3.4 or Guppy (v2.1.3) that have very similar read accuracy and consensus accuracy compared with albacore v2.2 used in our manuscript (Read accuracy: 88.09% vs 87.91% vs 87.15%, and consensus accuracy: 99.37%, 99.36%, 99.47% for albacore v2.2, albacore v.2.3.4 and Guppy v2.1.3 respectively).

Furthermore, Wierzbicki et al (2020) also concluded that Canu (that we used in this study) is the best assembler when it comes to TE quality metrics (TE abundance, TE SNPs, and TE internal deletions).

Still, following the reviewer's advice, we re-analyzed the long-read sequencing data of the *D. melanogaster* reference genome provided in Solares et al. 2018 by using the same pipeline we applied in our genomes (we first performed de novo genome assembly of the long-reads followed by polishing using short-reads). The obtained assembled genome was comparable in quality with the 32 genomes sequenced in our work (Table 1): 147.8Mb, 518 contigs, N50=3.4Mb, BUSCO C=96.0%. After scaffolding and masking the heterochromatic regions, we annotated transposable elements in the same way we previously did for our 32 genomes. We identified 2,215 TE copies, similar to what we have previously obtained (Table S9). In order to characterize the sequence accuracy of TEs when comparing with the reference ISO1 genome, we first identified orthologous TE copies between the two genomes using the same transference strategy applied in our work. We confidently determined 1,842 orthologous insertions, which represents >83% of the TEs annotated in Solares's genome and > 89% of the TEs annotated in the ISO1 reference genome. Then, for every TE pair we performed global pairwise alignments using MAFFT v.7.4 aligner (parameters: `mafft --globalpair --thread 4 --reorder --adjustdirection --auto`). For each pair we then calculated the pairwise identity in two ways: considering and not considering gaps in the alignment. Average gap-ignorant identity was 99.9% and gap-aware identity was 98.9%. Some families showed more variability than others but in most cases this variability was explained by individual TE insertions.

Thus, overall, we found no significant differences at the single nucleotide level when comparing TE annotations from the reference ISO1 genome and the TE annotations from a Nanopore-sequenced genome, which shows the high accuracy not only of the genomes but also the TE annotation obtained in the present work.

As suggested by the reviewer, we have included this new analysis in Table 1 and we have also added the details on the TE sequence accuracy in Supplementary Table S4 .

On a related note, I found the scaffold CUSCO [5] scores (p.5 lines 122-128) to be a questionable metric for quantifying the quality of TE sequences. To me, the CUSCO metric makes sense when computed for contigs: then, it is an assessment of how well some particularly challenging regions were assembled. However, the authors performed reference-based scaffolding for all their assemblies and reported a scaffold CUSCO (sc.CUSCO) score. In this context, the sc.CUSCO score is more of a measurement of how well the unique sequences flanking piRNA clusters are assembled and if they map to the reference, but also depend on how well piRNA clusters are assembled in the reference. The challenging sequences in the piRNA cluster do not need to be contiguously assembled to obtain a good sc.CUSCO score. I would speculate that if the flanking sequences are sufficiently complex and unique enough to be assembled with just short reads, a reference-scaffolded short-read assembly would also have a very high sc.CUSCO score.

>>>We agree with the reviewer. CUSCO score was recently described by Wierzbicki et al (2020). Wierzbicki et al. claimed that although the CUSCO metric calculated on contigs might be more biologically relevant than the scaffold CUSCO, estimating the two variables provides a more complete picture of the assembly quality. We are thus following the authors recommendations and we calculated both the contig and the scaffold values that are provided in Table 1 of our manuscript and mentioned in the main text. We have edited our manuscript to make this more clear, as our previous writing could have been misleading, and we agree with the reviewer observations.

2. Performance of long-read versus short-read methods (specifically, page 9 lines 237-299).

I thought this section was very confusing to read. At first glance, the header suggests that annotations based on long-read genomes (REPET) significantly outperform short-read methods (TIDAL, TEMP) [6,7] by 60%, but throughout the section, the numbers seem quite similar and it is suggested that the haploid genome used for the REPET approach is also limiting. If you take the numbers presented in the last paragraph then you get ~60%, but that's conditioning only on the 92 TE insertions uniquely detected by REPET out of the thousands that were detected.

>>>The number of insertions that were manually inspected were 132 that represents 1,452 TE calls. The 92 insertions mentioned by the reviewer are the subset of these 132 insertions that were detected by REPET and that allowed us to estimate how many of the insertions were missed by the other tools. We agree that this number is not too high and in this new version of the manuscript we have increased the number of insertions that were manually inspected to 350 TEs.

Briefly, we manually inspected 300 TEs that were annotated in the COR-014 genome by REPET and that allows us to estimate the false negative rates of TEMP and TIDAL and the false positive rate of REPET. Note that this number of TEs represents 42% of all the novo insertions detected in COR-014 genome, and as such we think it allows us to get a good

estimate of the false negative and false positive rates. Additionally, we manually inspected 50 of the insertions detected by TEMP/TIDAL but not by REPET. These 50 insertions were manually inspected in each one of the genomes analyzed to try to estimate REPET false negative rate.

In summary, in this new version of the manuscript, we increased the number of manually inspected TE insertions from 132 to 350 TE insertions, which is higher compared with other studies (e.g. 267 SVs in Chakraborty et al 2019) and we have rewritten this section to improve readability.

Another thing that was unclear was how the authors performed the comparison against TIDAL and TEMP [6,7]. Were the databases for those two programs the original databases from the TIDAL and TEMP studies? If the REPET library was built with the same genomes that TE identification was performed in, but the TIDAL and TEMP libraries were built from different samples, wouldn't REPET naturally perform better than TIDAL or TEMP in this scenario?

>>>In the first version of our manuscript, we followed the developers protocol to run TEMP and TIDAL. As such we used the RepBase library to run TEMP, and both the RepBase and Flybase libraries were used to run TIDAL. To compare the performance of the three software, we then focused only on insertions that belong to TE families that are common across the three software as reported in supplementary table S8A. Thus, the results that we were providing were comparable.

For this revised version of the manuscript, we have decided to re-run TIDAL and TEMP using the same TE library as for REPET. As such, we have increased the number of families that can be compared from 115 to 146 (all the families in the genome). To improve clarity, we now present the results separately for the comparison between REPET and the other two tools, and we report the results by genome. Besides identifying the TEs that are missed by the short read tools (false negatives of TEMP and of TIDAL), we also identified the false positives of REPET. To do this, we identify in one of our assembled genomes (COR-014) the region where the TE is annotated according to REPET/TEMP/TIDAL and we blast this region (i) against a database that contains all the TE sequences identified in our genomes and (ii) against the Flybase TE database. If we identify a TE but this TE is not annotated by TEMP/TIDAL we considered it a TEMP/TIDAL false negative. If a TE is annotated by REPET but the Blast does not confirm the presence of a insertion, we considered it a REPET false positive. Finally, as mentioned above, we also manually inspected 50 TEs annotated by TEMP/TIDAL but not REPET to estimate REPET false negative rate.

As mentioned above, we have rewritten this section of the manuscript.

This does not seem to strongly support the claim that the authors' approach "outperforms state-of-the-art methods for TE annotation using short-reads" (page 16, lines 483-484). I think that significant clarification to the methods and/or approaches to make this a more similar comparison (i.e. ensuring the same strains were used to build TE libraries across all methods) is necessary to demonstrate that this is so.

>>>See comment above. The results are directly comparable.

|

|

I also did not like that the terminology for the frequency classes (unique, polymorphic, fixed) was not defined before its usage nor in the Figure 4 caption.

>>>Thanks for pointing this out to us. As mentioned above, and to simplify this section, we now provide all the results by genome.

3. Characterization of TE population variability (specifically, page 12 lines 346-394)

I thought the way population variability in TEs is characterized could be better connected to the demographic history of *D. melanogaster*. Currently, this manuscript characterizes the counts of TEs in certain frequency classes (Figure 6) and gives some clues into the geographic distribution of TEs but does not provide any additional insight into these patterns.

One thing that immediately comes to mind is how the patterns of TE variability are similar or different to the patterns of SNPs across *D. melanogaster*. The consistent linear increase in the number of rare annotated TEs in Figure 6A seems to be consistent with the star-shaped genealogy of a recently expanded population, but does purifying selection on TEs also contribute to this? How does the frequency spectrum of TEs compare to nonsynonymous SNPs, synonymous SNPs, SNPs in conserved regions, “neutral” SNPs, etc.?

I will acknowledge that this is a big ask and these sorts of analyses may not be straightforward and outside of the scope of a paper revision/more appropriate for a new paper. However, the authors have emphasized the importance of understanding TE variability in natural populations in this manuscript. Providing at least a simple link between TE diversity and the population genetic forces in *D. melanogaster* (typically studied with SNPs) would help make this point.

>>> Following the reviewer’s suggestions, we analyzed the frequency spectrum of different classes of SNPs and compared them with the frequency spectrum of TEs. First, we classified the 2,785,420 previously called SNPs in three categories using the effect prediction tool SnpEff (v.4.3t):

HIGH: The SNP is assumed to have a high (disruptive) impact on the protein.

MODERATE: A non-disruptive SNP that might change protein effectiveness.

LOW: A SNP that is assumed to be mostly harmless or unlikely to change protein behaviour.

Since SNPs might have more than one effect classification depending on whether they overlap with more than one gene/transcripts, we applied the following criteria: HIGH impact variant if at least one effect is HIGH; MODERATE impact variant if at least one effect is MODERATE, but no HIGH effect is found in any other gene/transcript. LOW impact variant if only LOW impacts are predicted for the SNP. Besides, we also include another category: “Neutral SNPs”, containing SNPs located within the first 8–30 base pairs of small introns (≤ 65 bp) which are considered to be neutrally evolving (Parsch et al. 2010). Note that these neutral SNPs were also used for the selection analysis section of the manuscript.

As expected, the SFS for TEs showed an excess of rare variants consistent with purifying selection, as has been previously described (see e.g. Cridland et al 2013 and Bourgeois and Boissinot 2019 for a review). The SFS for TEs was more similar to HIGH and

MODERATE impact SNPs than to LOW impact and Neutral SNPs. Pairwise Kolmogorov-Smirnov Test for SFS distributions for different types of variants showed significant differences for all comparisons ($p < 0.0001$) (Supplementary Figure 11). However, D statistics showed greater differences between TEs vs. LOW impact and TEs vs. Neutral SNPs than TEs vs. HIGH/MODERATE impact SNPs, which is also consistent with a deleterious or slightly deleterious effect of TEs. While TEs, and SNPs with HIGH and MODERATE impact show a high proportion of variants at very low frequencies, a pattern consistent with the action of deleterious or slightly deleterious mutations that have not been yet removed from populations by purifying selection, LOW impact and Neutral SNPs show a much reduced proportion of low frequency variants, indicating that they tend to be more spread in the population due to their less -or even or null- negative effect.

Although we now mentioned in our manuscript that the SFS of TEs shows an excess of low frequency variants and have added an additional supplementary figure with this analysis (Supplementary Figure S11), elucidating how the different dynamics of TEs and SNPs affect the frequency distribution of these two types of variants is beyond the scope of this paper.

4. The role of TEs in adaptation (specifically p14-15 lines 429-462)

I don't know if the authors have convincingly demonstrated that these TEs are candidates for adaptation.

The authors took a set of 479 TEs at $>10\%$ frequency, performed selection scans in flanking regions, and found that 34 of these TEs were associated with a signal of positive selection. The methodology is a bit unclear (p.23-24, lines 685-709). Was the calculation of the statistics only performed with haplotypes with TEs present as lines 708-709 seems to imply? If not properly stratified across frequency classes, wouldn't the variance of these statistics differ across different TE frequencies, meaning the tails might be enriched for data points from a low frequency TE/high variance distribution?

Either way, a null expectation for the association between TEs and signatures of adaptation is not provided, so it is difficult to distinguish whether this is simply an overlap between two random, independent processes or whether a correlation (or a causal link) exists. *D. melanogaster* is notorious as a species in which adaptation occurs frequently. If one were to randomly choose 479 SNPs/loci in the *D. melanogaster* genome (ideally, matching for background characteristics like allele frequency, recombination rate, or the density of functional elements) and perform a similar scan, what proportion of these loci would you expect to see a signature of adaptation associated with? What would the enrichment in GO terms look like for the null sets?

Given that the role of TEs in adaptation is a central claim of this paper (it's in the title), a stronger link between TE insertions and adaptation needs to be established.

>>>All the details on how the analysis were performed were given in the Supplementary File 1. Unfortunately, this supplementary file was not made available to the reviewers. In the previous version of our manuscript, and because we could not phase TE insertions using long-reads, we were looking for evidence of selection considering that the TE insertions were homozygous and considering that they were heterozygous. We stratified our analysis by frequency classes and we compared with a null expectation obtained from

neutral SNPs. When revising the manuscript, we realized that this approach was not taking into account all the possible combinations of TE presence/absence in each one of the strains analyzed. Thus, in this new version of the manuscript, we have used a slightly different approach. Briefly, we looked for evidence of selection in SNPs stratified across frequency classes and using as a null expectation the statistics obtained for neutral SNPs. Once the SNPs under selection were identified, we identified TEs that co-occur with these SNPs and test whether candidate adaptive TEs (those present at high frequencies (>10% frequency and <95% frequency) in regions of the genome with high recombination rates (746 TEs) were more often associated with SNPs under selection compared with all TEs in the genome present at >5% frequency (2,697 TEs), as this is the minimum allele frequency required to test for SNPs under selection. Among the 2,697 TEs at freq >0.05, we found 40 TEs co-occurring with significant SNPs (1.48%), which was significantly lower than the 19/746 (2.54%) found among the candidate adaptive TE set (Chi-square = 7.012, p-value = 0.0081).

Although this analysis suggests that candidate adaptive TEs are more often associated with significant SNPs than all TEs in the genome, we acknowledge that these results are only suggestive but not conclusive evidence for the adaptive role of these TE insertions. We have re-writing this section of the manuscript to make these results and its limitations more clear. Note that one of the 18 TEs that are identified now as candidates to play a role in adaptive evolution is the well-known *Accord* insertion that has been shown to be involved in insecticide resistance (Daborn et al 2002). Besides, another of the 18 TEs was associated with variability in the level of expression of its nearby gene in nonstress conditions. We have included the expression of this TE in a new figure (Figure 6) together with the expression of the TEs with the most significant p-values in our eQTL analysis. Overall, we are confident that our analysis identifies *bona fide* candidate adaptive TEs. Still, we have edited the title of the manuscript to reflect that at this point it is only associative evidence, and as mentioned in the manuscript further analysis would be needed to functionally validate these insertions.

5. The discussion is underdeveloped

I thought this section could be better developed. As it stands, the Discussion consists of brief summaries of the work (e.g., better TE annotations) or broad, sweeping statements (e.g., long reads are useful, studying TEs may provide new insights). I wish the authors would discuss some specific implications of their results, how future studies possibly with even longer reads might open up new avenues for analyses, how analyzing data across different climate zones or across different species using their approach would be insightful, etc.

>>>We thank the reviewer for their suggestions to improve the discussion. Regarding the specific implications of our results, we explicitly mention the advantage of having more than one reference genome on the discovery of TE variants. We also discuss that future availability of haplotype-resolved de novo assemblies would allow incorporating polymorphism in the analysis and longer-reads should further help elucidate complex regions. Finally, we specifically discuss why analyzing genomes across climate zones would be insightful.

Minor comments:

1. The authors repeatedly state that their samples are from “five climatic regions” throughout the manuscript, but do not provide any rationale for why this kind of sampling is important, or how climatic regions relate to any of their results.

>>>We now provide more explicit rationale as to why this is important.

2. Page 5 lines 118-119: Did differences in read length explain differences in genome size or TE content?

>>> We found no correlation between read length (N50 of the reads) and the size of the assembled genomes (coefficient of determination $R^2=0.045$), or TE content ($R^2=0.0765$). We have now added this analysis to our manuscript (Figure S1).

3. Figure 6 caption: please explain what Figure 6B shows rather than just saying it’s an “UpSetR plot.”

>>>We have modified Figure 6B legend (now Figure 5B).

4. Figure 7 caption: please fully write out what each library is rather than using notation like “Allele_Phenotypes_from_FlyBase_2017”

>>>We have modified Figure 7 legend

5. There are a number of minor typos throughout the manuscript.

>>>We have revised the whole manuscript to try to identify all the typos.

References:

1. Watson et al. 2019, <https://www.nature.com/articles/s41587-018-0004-z>
2. Koren et al. 2019, <https://www.nature.com/articles/s41587-018-0005-y>
3. https://nanoporetech.github.io/medaka/draft_origin.html
4. Solares et al. 2018, <https://academic.oup.com/g3journal/article/8/10/3143/6026978>
5. Wierzbicki et al. 2020, <https://www.biorxiv.org/content/10.1101/2020.03.27.011312v1.full>
6. Zhuang et al. 2014, <https://academic.oup.com/nar/article/42/11/6826/1431591>
7. Rahman et al. 2015, <https://academic.oup.com/nar/article/43/22/10655/1803983>

Reviewer #2 (Remarks to the Author):

This is an important paper that uses long read sequencing (PacBio and Oxford Nanopore) to assembled 32 D. Mel genomes and semi-manually (with the help of automatic pipelines) curated a library of transposable elements (TEs). This library is then annotated against all 32 + 15 (from published study) genomes generated with long read sequencing to identify fixed and polymorphic TEs. For those with RNA-Seq data, the polymorphic TEs are tested for association with expression of genes nearby (eQTL analysis). Finally, TEs are assessed for signature of positive selection. Overall, the paper is very well written, addresses a

significant gap in our understanding of genome structure variation in *Drosophila*, and produces a useful resource (the TE library, especially the novel ones).

I have a few comments, mainly regarding data presentation, to improve readability of the paper:

1) The saturation analysis (what I call Figure 6) could be improved or done differently. Apparently the classification (frequency based) is variable depending on what strains are used to identify the TEs. I would fix the classification based on the full data set, then perform the same analysis again to draw Figure 6.

>>> We have followed the reviewer suggestion and we now used a fixed classification based on the full data set to perform the analysis. We considered a TE to be rare when it is present in < 10% of the strains analyzed, we considered a TE to be common when is present in $\geq 10\%$ and $\leq 95\%$ of the strains analyzed, and we considered a TE to be fixed when it is present in $> 95\%$ of the strains analyzed. The supplementary Table S11 and Figure 6A (now Figure 5A) have been updated accordingly.

2) I'm worried about the fact that nearly all TEs associated with expression were from the new TEs identified in this study. The eQTL analysis of the paper lacks some details. I would suggest pick a few TEs (significant and randomly selected non-significant ones):

- a) draw gene structure (with axons) along with the insertion sites of the detected TEs
- b) superimpose wiggle grams on them
- c) also draw a scatter plot of the estimated expression versus TE presence/absence

I commend the authors for being very careful in the TE annotation work, but this part lacks a bit of rigor.

>>> To check if TEs are associated with differences in expression level, we focus on non-fixed TEs (as these are the only ones that can be associated with differences in the level of expression). From this initial set of TEs, we checked how many of them are reference and how many are *de novo*. We found that out of the total 13,518 TEs analyzed, 779 are reference insertion and 12,739 are *de novo*. Thus, 94% of the TEs analyzed are *de novo*. Among the significant TEs, 2 were reference insertions and 470 were *de novo* insertions. As noticed by the reviewer there is a depletion of reference TEs among the significant TEs (Chi-square p-value < 0.00001). We do think that this depletion is at least partly explained because most of the significant TE-gene association involved TE insertions that are present in 1 out of the 20 genomes analyzed (462/472). While 11,681/12,739 (91.6%) of the *de novo* insertions are present in 1/20 genomes, only 108/779 (13.9%) of the reference TEs are present in 1/20 genomes. Thus, if we take into account the total number of TEs *de novo* and reference and the percentage of these TEs that are only present in one of the strains analyzed, we would expect to find four insertions and we found two.

We have followed the reviewer's advice and we have randomly picked 12 significant TEs and 12 nonsignificant TEs from the same families and we have plotted the expression levels of the strains with and without the TE insertion. However, we have chosen to plot the results using the most common type of plots for eQTL analysis.

Significant TE insertions:

|

|

Non-significant TE insertions:

Differences of expression appear to be more conspicuous in the 12 randomly chosen significant TEs compared with the 12 nonsignificant ones. We thought it would be more relevant for the reader to provide these plots for the 13 most significant TE insertions that are provided in Table 2 of the manuscript. We have also included in this figure the expression for the TE associated with signatures of selection (Figure 6).

3) For the gene set enrichment analysis, a more detailed description is needed in the Methods. For example, what is the background gene set? What is considered significant? What kind of test is used?

>>>We agree with the reviewer that this is important and we have now specified in the material and methods section the background gene set. We considered as significant those clusters with an enrichment score >1.3 as recommended by the authors of the DAVID tool (Huang et al 2008, Nature Protocols). For FlyEnrichR results, we report only terms with adjusted p-value <0.05.

4) Title could be better. First, the title should reflect TE rather than SV. Second, the relationship with adaptive evolution is weak.

>>>We have followed the reviewer's suggestion. The title of the manuscript now reads: *Population-scale long-read sequencing uncovers transposable element contributing to gene expression variation and associated with adaptive signatures in Drosophila melanogaster*

Minor comments:

- Figure 6. Why as the number of randomly sampled strains increases (from top to bottom, 10->47), the number of identified TEs decreases? Shouldn't it be the opposite trend?

>>>The number of identified TEs does not decrease. This figure shows the number of TEs classified as common *i.e.* present in $\geq 10\%$ and $\leq 95\%$ of the strains when analyzing only 10 strains, 20, 30, 40 and the whole 47 strains dataset. The number of TEs classified as common varies in these different datasets because the total number of strains analyzed changes and so thus the number of TEs present in $\geq 10\%$ and $\leq 95\%$ of the strains. We have revised the figure legend for this plot (now Figure 5).

- Line 98: please in this section note that 2 strains were sequenced using PacBio and 30 by MinION. This is an important piece of information that should be in the main text too.

>>>We agree with the reviewer. This information was in the material and methods section of the manuscript. We have now also added it to the legend of Table 1.

- Line 540: accessed -> assessed

>>>Thanks for pointing this out. We have now corrected it.

- Figure S3 title: mayor -> major, also I would call these dot plots rather than mummer plots.

>>>Thanks for pointing this out. We have now corrected it.

- On my reviewer page, I cannot find any of the supplemental files. These should somehow be made available.

>>>We apologize that the supplementary files were not available. We have made sure that they are available in the new submission.

Reviewer #3 (Remarks to the Author):

Repetitive sequences in the genome play an important role in genome and phenotypic evolution, yet much of them are inaccessible to the short reads. This limitation has impeded systematic functional and evolutionary analysis of the repetitive genomic sequences. In "Hidden intraspecies structural variation contributes to adaptive evolution in

Drosophila melanogaster” Rech et al. report de novo genome assembly of 32 *D. melanogaster* strains from Europe and North America. They assembled genomes using Oxford Nanopore long reads and then created a custom repeat library to comprehensively map the transposable elements in the assembled genomes. Their analysis yielded 3 new TEs in the well-studied model organism *D. melanogaster*. They further showed that short reads based methods of TE annotation fail to detect 60% of the TEs they found in the assemblies. Among the TEs they detected in their assemblies, several are located in genomic regions showing signatures of positive selection.

Finally, they show that >400 TEs are associated with gene expression variation, suggesting that TEs are a potential source of gene regulatory variation.

The 32 new de novo genome assemblies reported by the paper will be a useful resource for the *Drosophila* community and population geneticists in general. The result on novel TEs is interesting and so are the newly uncovered putatively adaptive TEs. However, many details and citations of prior work in the relevant sections are missing, making it harder to evaluate the claims by the authors about the discovery of new TEs and their adaptive and functional significance. The paper would benefit greatly from providing these details and clarifications. My detailed comments are below.

>>>We thank the reviewer for their suggestions that we do think have helped improve our manuscript. We would also like to apologize for failing to cite some prior work. We are aware of all the publications mentioned by the reviewer and we did not compare our results to their results as we thought they were not directly comparable (please see below). Still, we have now added these comparisons in the new version of our manuscript.

The criteria for calling a TE hidden vs visible was unclear. The authors need to clearly mention the criteria used for hidden vs visible/novel TEs in all contexts they are mentioned in the paper (e.g. short reads vs assembly comparison, putatively adaptive TEs, putative cis-regulatory TEs). Are TEs being described as variants or just genomic elements? In some sections, authors describe the TEs as genomic elements (TE annotation) but in other sections they are treated as variants (e.g. putative adaptive variants). Were the new TE families identified by the authors present in the 14 reference *D. melanogaster* genomes previously sequenced (doi: 10.1038/s41467-019-12884-1)? Were the variants hidden because they escaped detection by TE genotyping pipelines based on paired-end short reads (e.g. PMID 23883524) ? Due to this lack of clarity, it was not immediately clear whether novel TEs described in the sections on signature of adaptation and gene expression are novel because they could not be detected with short reads or because these genomes were not studied before. At the very least, the authors should support their claims about novel TE variants by comparing their calls to a short reads based genotyping pipeline (e.g. PMID 23883524) and show that those TEs were invisible to the short reads.

>>>We apologize if it was not clear what hidden meant. The same criteria was applied in the whole paper. Briefly, all TE insertions that were not previously annotated are considered “hidden”. These previously non-annotated TEs are a combination of TEs that were not annotated because the previous library could not identify them, and TEs that were not annotated because those particular genomes have never been studied before. We have now modified the title to avoid this word and thus improve clarity. We have also revised the manuscript to avoid inconsistencies in the terminology.

Regarding whether the new TE families were present in the 14 reference genomes previously sequenced, we would like to point out that Table S10 shows in which genomes were present each one of the TEs annotated in this work. All three families are present in the 14 reference genomes previously sequenced, and the copy number in these genomes is similar to the one in other genomes analyzed. We provide the number of copies of the three families identified in each one of the 14 genomes below but again, this information can be extracted from Table S10 of the manuscript.

Genome naem	NewFam06	NewFam14	NewFam16
A1	17	8	4
A2	14	7	3
A3	15	7	3
A4	17	7	4
A5	19	8	4
A6	14	7	4
A7	20	8	3
AB8	15	10	3
B1	18	9	3
B2	18	8	3
B3	18	9	4
B4	14	8	4
B6	16	8	4
ORE	18	7	3

Finally, and as mentioned above, hidden TEs are novel because they could not be detected with the previous library (that did not contain all the copies of the new genomes reported in this work) and also because short reads failed to detect a substantial proportion of TE insertions compared with long-reads. In the previous version of the manuscript we already showed that annotation with our library discovers 58% more insertions compared with the current annotation of the reference genome. In this new version, and to provide further evidence of the increase in the number of copies annotated thanks to our more complete TE library, we have used Repeatmasker to annotate our core 13 genomes with the previously available BDGP library and with the new library constructed in this work. We found that the BDGP library does not annotate 42% to 44% of the TE copies annotated with the MCTE library. We have added this new analysis to our manuscript (Supp Table S6D).

In the previous version of the manuscript we also already showed that a substantial proportion of the insertions escape detection based on short reads. In this new version we have further improved this section by considering all the TE families in the new library rather than the ones overlapping with previous TE libraries (please see response to reviewer above and the revised section in the manuscript).

The method of detecting candidate adaptive TEs was a bit unclear. The Methods section says that the authors used SNPs for the tests of selection, but in the main text they present the TEs as the putative adaptive variants. How did the authors narrow down their search from a collection of SNPs to a particular TE? Were the TEs encoded as variants in their analysis and TE variants showed zero LD with the adjacent SNPs? This needs to be clarified because this is one of the major results of their paper. I am particularly curious about how the authors disentangled the candidate TEs from linked SNPs or other small or large SVs at the focal regions showing the signatures of positive selection. The claim of potential adaptive significance of the TEs would be strengthened if the authors provided the gene expression phenotypes of their candidate adaptive TEs.

>>>We apologize that this was not clearly explained in the first version of our manuscript. All the details were included in a supplementary file that was not made available to the reviewers. We have revised the methodology used to look for evidence of selection and as such we have now re-written this section of the manuscript. We do require the TE to be in linkage disequilibrium with the SNP. In this new version of the manuscript, we have also discarded the presence of other SVs in these regions. We have run SVMU, a computational tool that identifies structural variants, in our genomes (Chakraborty et al 2019). We then focused on the regions around the 18 TE insertions found to be associated with signatures of positive selection and identified the CNVs present in 1kb regions upstream and downstream of each insertion. We found that five of the 18 candidate adaptive insertions are located in genomic regions where additional CNVs have been detected. However, in any case the CNVs were in linkage disequilibrium with the TEs. We have added these new results to our manuscript (Supp Table S14).

Finally, have also plotted the gene expression phenotypes for the candidate adaptive TEs.

However, we have decided to only include in our manuscript the expression of the TE that showed significant association with expression level variation (new Fig 6).

The authors missed citation of prior work in several sections. Citing prior work would not only acknowledge the work people have already done, but will also place the results described in the manuscript in a larger context, strengthening the claims of novelty. One prominent omission was the citation of the work Julie Cridland did on showing the association between TEs and gene expression in DGRP (doi: 10.1534/genetics.114.170837). Did the authors sequence any of the lines Cridland et al. used in their study? If they did, it would be interesting to know whether any of the variants that Cridland et al. missed in the DGRP strains was detected by the new assemblies and showed association with gene expression variation. Similarly, the extent to which TE insertions are missed by short reads in *D. melanogaster* was first quantified and reported by Chakraborty et al. (doi: 10.1038/s41588-017-0010-y). On a related note, a high frequency copy number variant (deletion) of *Cyp6a17*, showing a large effect on temperature preference behavior, has been reported previously (doi: 10.1038/s41588-017-0010-y). I am curious about the status and role of this CNV in the gene expression variation the authors observe for *Cyp6a17*. Finally, I was wondering about why the authors do not cite any paper from *D. melanogaster* that has been published in the last five years when they discuss the limitations of short reads in discovery of genetic variation due to repetitive sequences in the first paragraph of Introduction.

>>>We completely agree with the reviewer and we apologize for the oversight. We are well-aware of all the works mentioned by the reviewer and we have now fixed this in the revised version of our manuscript. Thanks for pointing this out to us in such a constructive way.

We did initially not compare our results with Cridland et al's results as they focused on the effect on expression of rare TE insertions while we evaluated all TE insertions (independently of their population frequency). There is only one strain in common between our dataset and that of Cridland et al (RAL-375), which analyzed 37 inbred RAL strains. In total, 282 non-reference TE insertions were detected in the RAL-375 strain according to Cridland et al. By using bedtools intersect we detect that ten of these TE insertions were located in heterochromatin regions, so we discarded them from the following analysis as our analysis focuses on euchromatic regions. In order to test how many of the 272 TE insertions were detected in our analysis, (we identified 691 non-reference TEs in this strain) we expand the insertion site by ten nucleotides to allow for small inconsistencies in the exact insertion site coordinate and intersect both files using bedtools intersect. We found that 232/272 (85%) of the TEs identified by Cridland were also found in our dataset. Then we check how many of the non-reference TEs identified based on our analysis (691 TEs) were also founded by Cridland, and found that 232/691 (35.6%) of them were common to both datasets. Thus, while we identify the majority of the TE insertions reported in Cridland (85%), the use of short reads missed the majority (66.4%) of TE insertions identified in our work. Moreover, 20 of the TEs missed by Cridland et al show differences of expression in our work. We could not check the overlap of the changes in expression between the two works because the data is not available for Cridland et al analyses. We now discussed Cridland et al results in our manuscript.

The reviewer is correct that Chakraborty et al 2018, and also Chakraborty et al 2019, compared their TE annotations with the annotations based on short-reads. We did not include this in our manuscript as we thought the annotations performed by Charkraborty and our annotations were not directly comparable. Charkraborty et al (2018) first annotates structural variants and then assigns indels to TE insertion based on sequence homology using available TE libraries with the Repeatmasker tool. This approach would be similar to running annotations with the available *Drosophila* TE library rather than creating a new one that incorporates the TE variation present in the natural populations studied in our work. Charkraborty et al report that a previous annotation of the A4 strain based on short-read sequencing fails to find 38% of the TE insertions they annotate based on short reads. Chakraborty et al 2019 reports this percentage to be 36% based on the analysis of 14 genomes. As mentioned above, we found that 26-67% of the TE insertions are missed by short-read sequencing. We have now added the previous available estimates to our manuscript.

Regarding the *Cyp6a17*, this information was included in the previous version of our manuscript. All the strains with the TE insertion had also the triplication of the gene, as such the effect of the two events cannot be separated. Finally, we now also have added papers based on *D. melanogaster* in the first paragraph of the introduction. In the first version of the manuscript, for this first introduction paragraph,

we decided to focus on organisms other than *Drosophila* but we agree with the reviewer and we have now included *Drosophila* papers as well. Thanks for the recommendation.

The site frequency spectrum of TEs suggest that on average TE insertions are strongly deleterious in *D. melanogaster* (doi: 10.1038/s41467-019-12884-1). It is therefore expected that most polymorphic TEs are rare and the vast majority of the TE variants will not be shared by individuals. Therefore, the motivation for the section on most common variants can be sampled by 20 individuals is not clear. In this section, I think the authors meant to say ‘only additional 36’ instead of “only 36” at Line 374. Same goes for “27 TEs” at line 375. I thought the most interesting part in this section was the TEs showing evidence for their roles in local adaptation, but relatively less real estate is spent on this result.

>>>We have tried to clarify our motivation for this section. The reviewer is correct that most polymorphic TEs are rare and the vast majority of TE variants will not be shared by individuals. However, the question of how many genomes are needed to discover most of the TE variants that are common in *D. melanogaster* natural populations was not answered yet, and we think is relevant as it shows the minimum amount of genomes needed to get a good sampling of common TE variants which is relevant for example if you want to identify candidate adaptive mutations.

The reviewer is correct and we have now added “additional” to our statements. We have now performed a DAVID analysis with the genes located nearby the TEs that are present at high frequencies in only one of the two continents. However, the number of genes is too small, and we did not identify any enriched biological processes. We have added this new information to our manuscript.

Please provide the error rate of the assemblies at the nucleotide level (QV). Details on how to estimate QVs can be found in Solares et al. (doi: 10.1534/g3.118.200162).

>>> We calculated the QV scores following Solares et al 2018 for the 30 assembled genomes that were sequenced with ONT, and thus were polished with short-reads, and with the reference genome. Briefly, we aligned Illumina short-reads to each of the assembled genomes and we then performed SNPs and INDELS calling using GATK best practices pipeline. Then we calculated the *Perror* as $Perror = Variants / (Total\ Bases)$, where “Total bases” represents the total number of aligned bases. Finally, QV score for each assembly was determined using the formula: $QV = -10 * \log_{10} Perror$. We calculated two different QV scores, one considering only SNPs variants and other considering both SNPs+INDELS variants. QV-SNPs scores ranged from 37.2 to 59.2 (mean 44.2) and QV-SNPs+INDELS ranged from 36.0 to 51.4 (mean 42.4) . Both scores resulted in similar values to the previously obtained by Solares et al. 2018, suggesting that our assemblies are of high quality in areas where reads align well. We have added this information to our manuscript.

The title of the paper is misleading. The authors do not show conclusively that TEs are the causal adaptive variants, making the claim about newly discovered TEs contributing to adaptation only speculative. Additionally, the authors discuss only one class of SVs (TE insertions) in the paper, but the title gives the impression that the paper is about all classes of SVs.

>>>We have edited the title of our manuscript following the reviewers suggestions.

|

|

It was surprising that reference-guided scaffolding drastically improved the CUSCO scores of the assemblies (64.1% to 93.7%). Surprising because scaffolding does not really add any new sequences to bridge gaps (or resolve repeats) in the assemblies. Could the authors comment on why scaffolding improved the CUSCO scores so drastically?

>>> We have revised this section of the manuscript following the reviewers comments. We think scaffolding improved the CUSCO scores because it is based on how well the flanking regions of the piRNA clusters are represented in our genomes.

Does RepeatMasker find any hit based on the latest release of the Drosophila TE library at the genomic regions where the newly discovered TIR element is found? Or, does it report nothing in those regions?

>>> To answer the reviewer question, we have analyzed 5 genomes (ISO-1, AKA-017, MUN-016, RAL-375, and TOM-08) using RepeatMasker v4.1.2-p1 (*-nolow* option) with **1)** RMBLAST search engine over the last RepBase library rb20181026 (RepBase RepeatMasker Edition, final version 10/26/2018) and **2)** HMMER search engine over the last Dfam library (Version 3.3, Date: 2020-11-09). In both cases we used the TE libraries for the whole "Drosophilidae" group. Then, we converted RepeatMasker results to bed format and used Bedtools intersect to find overlaps between REPET annotations and RepeatMasker. In these 5 genomes, which according to our annotation contain 13 to 19 copies of the TIR element per genome, Repeatmasker does not identify any interspersed repeats where the newly discovered TIR elements were found.

The status of heterozygosity in the strains used in the expression GWAS experiment is not clearly stated. Were these strains inbred? If they are not inbred, did the authors take the heterozygosity of the strains into account while doing the association study? A related question is: how did the authors perform haplotype based selection tests in heterozygous strains, where the number of segregating chromosomes are unknown?

>>> Most strains used in the study are inbred stocks from natural populations, with the exception of five strains (AKA-018, MUN-020, SLA-001, STO-022, TEN-015), which are isofemale strains. We provide detailed information on the origin and inbred status of each strain in Table S1. We evaluated levels of heterozygosity in all sequenced strains using short-reads sequences, which are summarized in Table S2C. High levels of heterozygosity in sequenced organisms is a well-known problem in genome assembly since assemblers tend to create a duplicated region in the heterozygous loci (one for each allele) instead of collapsing the whole region into one single haplotype. Because of this, as part of this work we deduplicate the *de novo* genome assembly prior to scaffolding. To do this, we applied a bioinformatic strategy (called *purge_haplotigs*, DOI:10.1186/s12859-018-2485-7) to identify and remove allelic variants in the primary assembly (predominant haplotype). After this processing we expect genomes to be a haploid representation of each strain. This was explained in the Material and Methods section of our manuscript and further details were given in Supplementary File 1.

All subsequent analyses were performed in these haploid genomes, and as such this was the assumption in all tests. In the specific case of selection tests, we have changed the strategy to identify selection and a detailed explanation is given in Supplementary File S1.12. Briefly, we require the SNPs under selection and the candidate adaptive TE to co-

occur in the same strain to consider that the regions flanking the adaptive TEs show signatures of positive selection. Still, we are well aware that this evidence is only suggestive but by no means conclusive of the TE being selected and we make this clear in our manuscript.

Reviewer #4 (Remarks to the Author):

Illumina and other short read technologies are not ideal for inferences about Transposable Elements, and for this reason this group decided to apply long-read sequencing methods on a set of 32 *D. melanogaster* strains. The project is well motivated and results in 32 high quality genome sequences of *Drosophila melanogaster* from 12 geographical populations. The authors applied Oxford Nanopore technology for 30 of the genomes, and PacBio for 2 of them. The procedures all appear to be state of the art, and the assembly procedures all reasonable. In the end there was a huge range of N50 statistics, going from 400 kb to 19 Mb. The authors claim that this seemed to make little difference to BUSCO scores or TE call accuracy, but the latter seems a bit anecdotal. Much of the analysis centers of inference of Transposable Element inference, an aspect of genome assemblies where short reads are particularly poor, and this problem is largely resolved by both long read technologies. The paper provides a solid technical presentation of the genome assemblies and inferences of TE insertions, including inference of effects on altered gene expression and detection of positive selection. A few specific technical issues follows:

>>>We thank the reviewer for their comments on our manuscript. In the previous version, we showed that differences in sequencing coverage across genomes did not explain differences in raw assembly sizes in MB, number of TE copies annotated and percentage of the genome corresponding to TEs (main text and supplementary Figure S1). In this revised version, and following the suggestion of reviewer #1, we also showed that differences in read length do not explain differences in genome size or TE content. Additionally, we have also tested whether N50 correlates with BUSCO scores ($R^2=0.0004$), TE copy number (0.2245) or TE genome content (0.0479). These new analysis have been added to the manuscript.

1. Regarding the calling of TEs, it is probably worth running RepeatModeler2 as well. Claims in Flynn et al. (PNAS 2020) make it appear to perform very well.

>>>Rodriguez and Makalowski (2021) have recently performed an evaluation of software to *de novo* detect TE insertions. They evaluated RepeatScout, REPET and RepeatModeler. The number of TE models representing unique TE families identified by REPET and RepeatModeler were very similar, 82 and 80 respectively for simulated data (the real number was 20), and 557 and 686 respectively for real data (*D. melanogaster* ISO-1: 126 families described). They conclude that extensive manual curation is needed with any of the tools used as they tend to identify TE-regions in a fragmented manner. Thus REPET performance is similar to RepeatModeler and both require manual curation.

We have indeed run RepeatModeler and REPET in the reference genome and obtained similar results to Rodriguez and Makalowski: both softwares identify a high number of consensus sequences and both require extensive manual curation to arrive to a good annotation. We do think this is beyond the scope of the paper and we have not include it in our manuscript. We could include it if the reviewer thinks is relevant.

2. Regarding the claim that the long read assemblies detect 58% more insertions, this reader wonders whether it would be useful to provide more definitive validation of these insertions, such as PCR. This does not seem to be the standard in this area, so it seems that confidence in the assemblies is very high, but surely at some point direct validation was done.

>>>We do not think a PCR will provide a more definitive validation of the insertions present in the reference genome. These 58% more insertions are regions of the genome that were not previously identified as TE insertions, because the library that was used to annotate the reference genome did not identify them as TE insertions.

For a subset of the insertions that we discovered in the 32 genomes and that are not present in the reference genome, we have visually inspected the alignments as a method to validate our annotations. This is considered to be a very accurate validation approach (De Coster et al 2021). These manual inspections suggest that the false discovery rate of REPET is only 3%. This information is now included in the manuscript.

3. In the section on inference of TE insertions associated with changes in gene expression, this reader is not sure whether the standard method of PCA covariates is adequate to control for population structure, especially when the population structure is as great as it is with *D. melanogaster*, and such small samples from so many populations is included. Some added justification seems warranted.

>>>We acknowledge that the strains that we selected for the eQTL analysis have a complex population structure. However, we did address this issue by incorporating the three first dimensions of the PCA in our analysis (Figure S21-S22), as commonly done in other articles using QTL-Tools and following the recommendation of the QTLTools authors that advise us to use the PCA covariates to control for population structure in our particular analysis (Dr Halit Ongen, personal communication). Figure S21 clearly shows a separation between American and European strains and this difference is taken into account when the covariate file is incorporated into the analysis.

4. Lines 237-299 A LOT of space devoted to comparison of calls of REPET, TIDAL and TEMP. This seems like material that could better be relegated to the Supplement. The main point that short reads miss 60% of TEs should be in main text.

>>>We have reduced this section of the manuscript, however we think that some details should be given so that the reader is convinced that the analysis is sound.

5. Lines 301-344 – The section discussing the similar numbers of TEs per line is rather qualitative, and should include well known population genetic metrics for heterogeneity among lines and among populations. Is there any theoretical expectation for these numbers, given ballpark estimates of rates of transposition.

>>>The reviewer is correct that we did not compare with previous studies reporting numbers of TE insertions across populations. This has now been fixed. We did however compare other metrics (e.g. TE order and TE family abundance) with previous results reported by Lerat et al 2019, that analyzed the TE content in 42 pooled-seq samples from European populations; Linheiro and Bergman 2012 that analyzed 166 individual strains;

|

|

and several other more specific works. We have also performed several statistical tests to back-up our claims.

Rates of transposition have previously been estimated at the superfamily level by Adrion et al 2017 by analyzing eight mutation accumulation lines. They identified 24 active superfamilies, however superfamily-specific insertion and deletion rates were not significantly different between lines.

6. Line 346-394: The focus in this section is on subsampling to see what is the sample size needed to capture most common TEs. The claim that 20 is sufficient again seems to lack quantitative rigor. Similarly, the count of population-specific TEs depends on their frequency spectrum and the sample sizes.

>>>We apply the Chi-square with Yates correction test to evaluate whether the % of common TEs is significantly different between subsets containing different numbers of strains (from 20 to 47 strains). We found that the number of common TEs detected using 42 strains (887 TEs detected as common) is already significantly different from the number of common TEs detected using the whole dataset (894 TEs; Chi squared p-value = 0.0066. However, the difference in the number of TEs detected is very small: from 887 to 894. We thus decided to check the standard deviation of the number of TEs identified as common when using increasing numbers of strains. The graph below shows that for common TEs when we analyzed 20 strains, the standard variation falls suggesting that 20 genomes is an accurate threshold to identify common TEs as the number of false positives (TEs that are classified as common when they are not) substantially diminishes.

7. Lines 396-427 center on functional effects associated with TE insertions, highlighting a role for TEs as cis-eQTLs. This comes across as mostly being anecdotes and not much new. Given this more exhaustive sample than before, can we make conclusions about the fraction of TE insertions expected to impact gene expression, whether different classes of TEs have different effects, and whether there is any signature of selection as a result of these effects?

>>>We respectfully disagree with the reviewer, and we would like to point out that other three reviewers have considered this as one of the relevant results of the manuscript. To the best of our knowledge there is only one previous work that assessed the role of TEs in gene expression variation genome-wide: Cridland et al 2015. This previous work analyzed 37 strains from a single population and focused on TEs present in only one strain (unique insertions). They conclude that unique TEs are associated with reductions of gene expression. We, on the other hand, analyzed both unique and polymorphic insertions. We identified 472 TEs significantly associated with the level of expression of the nearby gene. The majority of these insertions were present at low population frequencies (462/472), however, 214 were associated with gene up-regulation and 258 with gene down-regulation, while previous analysis based on a smaller dataset suggested that low frequency insertions are associated with gene down-regulation (Cridland et al 2015). We were able to analyze 12,283 TEs, thus 4% of the TEs show a significant association with changes in expression. However, note that our dataset for this analysis is based on 20 genomes. We did included in our manuscript the family enrichment analysis for these 472 TEs; we have now added the enrichment analysis considering the up- and down-regulated genes separately. Finally, one of the TEs that show signatures of selection is significantly associated with gene expression changes in our dataset. This TE is now depicted in Figure 6.

8. In Lines 429-462 the authors report that some 36 TEs in the set display evidence for positive selection, as assessed by tests identifying flanking haplotypes that appear to have expanded in frequency (e.g. the iHS test). These are awfully small sample sizes for inference of selection, so perhaps a word is warranted about the power of these tests, and the resulting conclusion that rather strong selection appears to be acting on these positive cases.

>>>We agree with the reviewer that the sample size is small. We have added a sentence along these lines.

A couple of trivial edits:

Line 469 change “methodical” to “methodological”

>>>Thanks for pointing this out. We have corrected it.

Line 692 change “mayor” to “major”

>>>Thanks for pointing this out. We have corrected it.

Reviewers' Comments:

Reviewer #1:

Remarks to the Author:

Overall, I think the authors have done a good job of responding to my concerns. I appreciate the additional genome QC and think this provides much more confidence in the quality of the genomes and their repeat content. I also find the tempered explanation of the relationship between TEs and adaptive evolution to be more acceptable in the context of the authors' findings.

There is one remaining issue that I'd like to encourage the authors to consider further. Overall, it's a minor issue, but I still find the presentation to be misleading.

I still don't think the authors are utilizing the CUSCO metric in its intended manner. In Wierzbicki et al. 2020, the authors emphasize an important distinction between ungapped and gapped CUSCO scores. The former does not tolerate any N sequences between flanking sequences; the latter tolerates Ns. This is mostly equivalent to the authors' contig and scaffold CUSCOs (respectively). The crucial difference is that Wierzbicki et al. are using gapped CUSCO to measure the effect of scaffolding with Hi-C, not reference-based scaffolding. In the case of this paper, genomes are being scaffolded with the reference so this metric isn't really informative except to say that the flanking regions exist in the de novo assembly.

My concern is that the sc.CUSCO scores create the illusion that the scaffolded assemblies are drastically improved in terms of repetitive content (as an extreme example, 35.29% to 92.94% for COR-023) when it is possible that the region is still very poorly assembled. In this case I think it's important to show other metrics like base and soft-clip coverage (Fig 2c/2d in Wierzbicki) for these regions. While I don't have an objection to reporting the sc.CUSCO scores in Table 1, it's the way the Results are written ("Scaffolding also significantly increased CUSCO scores...") that makes this misleading. Also note that any non-reference variability in these regions would be missed.

Other than this, there are a few small typos throughout the manuscript.

Reviewer #2:

Remarks to the Author:

This is a resubmission of a previously reviewed manuscript, which has improved substantially. The authors have successfully addressed all my concerns.

Reviewer #3:

Remarks to the Author:

The clarity and organization of the manuscript by Rech et al. has vastly improved compared to the earlier version. The most significant results of the paper are the 32 new reference genomes and a new custom TE library based on de novo repeat annotation. These resources will be extremely useful to the community. The new TEs in the new repeat library are particularly significant because the *D. melanogaster* reference TE library is considered gold standard. The role of TEs in gene expression variation is not a new finding by itself. However, in combination with the new TE library, the gene expression-TE association result bolsters the authors' claim about the value of using reference genomes for studying genome evolution and functional genetic variation. The role of newly annotated TEs in adaptations, as the authors point out, remains suggestive and will require substantial experimental work to conclude their true adaptive role and will be a major project. I only have minor comments:

1) The lower CUSCO score (64.1%) of the contig assemblies is a bit concerning, especially because the assembly and annotation of new TEs is a major result of the paper. To demonstrate the quality of their genome assemblies and TE annotations, the authors show that 85% of the assembled ISO-1 TE sequences map to the reference TEs with high identity. However, this number looks significantly lower compared to the initial PacBio assembly of the ISO1 strain in Berlin et al. (doi: 10.1038/nbt.3238) (Berlin et al. found 95% of the reference TEs aligning perfectly to the reference. supplementary note 7). Is this discrepancy (i.e. why Berlin et al. recover 95% and the authors of this study recover 85% of the reference TE) due to the differences in pipelines used in the two studies? For consistency, it would be nice if the authors run the pipeline Berlin et al. used and report the corresponding number for their ISO1 assembly.

2) In the Introduction, the authors state that they used strains from 5 different climatic regions. However, in the first paragraph of the Results section, they say 24 strains were collected from 11 geographic locations. For consistency and to avoid confusion, it might be best to explain the geographic regions of origin of the strains initially and be consistent about the numbers throughout the manuscript. For example, in the "A new manually curated...." section, the authors mention 12 geographic regions (is that 11 + North Carolina strains?).

3) The TE names in the X axis of Fig. 2A were difficult to read. I wonder if the font size could be increased a tad more?

REVIEWER COMMENTS

Reviewer #1 (Remarks to the Author):

Overall, I think the authors have done a good job of responding to my concerns. I appreciate the additional genome QC and think this provides much more confidence in the quality of the genomes and their repeat content. I also find the tempered explanation of the relationship between TEs and adaptive evolution to be more acceptable in the context of the authors' findings.

>>>We thank the reviewer for revising the new version of the manuscript

There is one remaining issue that I'd like to encourage the authors to consider further. Overall, it's a minor issue, but I still find the presentation to be misleading.

I still don't think the authors are utilizing the CUSCO metric in its intended manner. In Wierzbicki et al. 2020, the authors emphasize an important distinction between ungapped and gapped CUSCO scores. The former does not tolerate any N sequences between flanking sequences; the latter tolerates Ns. This is mostly equivalent to the authors' contig and scaffold CUSCOs (respectively). The crucial difference is that Wierzbicki et al. are using gapped CUSCO to measure the effect of scaffolding with Hi-C, not reference-based scaffolding. In the case of this paper, genomes are being scaffolded with the reference so this metric isn't really informative except to say that the flanking regions exist in the de novo assembly.

My concern is that the sc.CUSCO scores create the illusion that the scaffolded assemblies are drastically improved in terms of repetitive content (as an extreme example, 35.29% to 92.94% for COR-023) when it is possible that the region is still very poorly assembled. In this case I think it's important to show other metrics like base and soft-clip coverage (Fig 2c/2d in Wierzbicki) for these regions. While I don't have an objection to reporting the sc.CUSCO scores in Table 1, it's the way the Results are written ("Scaffolding also significantly increased CUSCO scores...") that makes this misleading. Also note that any non-reference variability in these regions would be missed.

>>>We thank the reviewer for this comment. We have revised the Wierzbicki paper and we agree with the reviewer that our mention to the scCUSCO as it was in the manuscript was misleading. We have revised these in our manuscript that now reads:

"CUSCO scores, *i.e.* percentage of contiguously assembled piRNA clusters (Wierzbicki et al 2020) range from 35.3% to 84.7 % (average 64.1%; Table 1). The detectability of a cluster was inversely correlated with its size (Pearson's correlation = -0.47; Table S3B, Figure S2 and Supplementary File S1.5)."

We have also modify Table 1 legend that now includes:

"Besides contig-CUSCO scores (c.CUSCO) scaffold-CUSCO scores (sc.CUSCO) are also given (the later values are higher as expected if the piRNA flanking regions are present in the assembled genomes."

Other than this, there are a few small typos throughout the manuscript.

>>>We have revised the manuscript and corrected several typos.

Reviewer #2 (Remarks to the Author):

This is a resubmission of a previously reviewed manuscript, which has improved substantially. The authors have successfully addressed all my concerns.

>>>We thank the reviewer for revising the new version of the manuscript

Reviewer #3 (Remarks to the Author):

The clarity and organization of the manuscript by Rech et al. has vastly improved compared to the earlier version. The most significant results of the paper are the 32 new reference genomes and a new custom TE library based on de novo repeat annotation. These resources will be extremely useful to the community. The new TEs in the new repeat library are particularly significant because the *D. melanogaster* reference TE library is considered gold standard. The role of TEs in gene expression variation is not a new finding by itself. However, in combination with the new TE library, the gene expression-TE association result bolsters the authors' claim about the value of using reference genomes for studying genome evolution and functional genetic variation. The role of newly annotated TEs in adaptations, as the authors point out, remains suggestive and will require substantial experimental work to conclude their true adaptive role and will be a major project. I only have minor comments:

>>>We thank the reviewer for revising the new version of the manuscript

1)The lower CUSCO score (64.1%) of the contig assemblies is a bit concerning, especially because the assembly and annotation of new TEs is a major result of the paper. To demonstrate the quality of their genome assemblies and TE annotations, the authors show that 85% of the assembled ISO-1 TE sequences map to the reference TEs with high identity. However, this number looks significantly lower compared to the initial PacBio assembly of the ISO1 strain in Berlin et al. (doi: 10.1038/nbt.3238) (Berline et al. found 95% of the reference TEs aligning perfectly to the reference. supplementary note 7). Is this discrepancy (i.e. why Berlin et al. recover 95% and the authors of this study recover 85% of the reference TE) due to the differences in pipelines used in the two studies? For consistency, it would be nice if the authors run the pipeline Berlin et al. used and report the corresponding number for their ISO1 assembly.

>>>We agree with the reviewer that genome assembly strategies and TE annotation pipelines may influence TE annotation. However, we first would like to clarify some points regarding this comment:

The reviewer mentions that we showed that "85% of the assembled ISO-1 TE sequences map to the reference TEs with high identity". The 85% mentioned in our manuscript is not based on the assembled ISO-1 nor on TE annotations performed by us, but is the result of comparing the annotations available in Flybase and the annotations based on REPET of the reference genome available in Flybase. We indeed also compared the annotation based on our assembly of the ISO-1 reads of Solares with the Flybase annotation, but in this case the overlap is 89% (as mentioned in the Methods section (page 21).

Moreover, this 89% is not entirely comparable with the 95% reported by Berlin et al, since the latter publication used the reference genome annotation while we used our annotation strategy based on REPET and the Manually curated TE (MCTE) library, whose consensus sequences also include the TE variability observed in *D. melanogaster* natural populations and not only in the ISO-1 reference.

Still, and as suggested by the reviewer, we applied the pipeline reported by Berlin et al. to both ISO-1 genomes, the one assembled by Berlin et al. and the one assembled by us using Solares et. al (2018) ONT data. The table below shows a summary of the results:

Total Number of TEs: 5,433	BERLIN et al	SOLARES et al
TEs identified	5,304	5,277
TEs identified (%)	97.63%	97.13%
TEs \geq 100% Pct_length & 100% Pct_ident	4,335	3,380
TEs \geq 100% Pct_length & 100% Pct_ident (%)	81.73%	64.05%

From the 5,433 TEs analyzed in Berlin et al we identified ~97% in both genomes (similar to the percentage reported by Berlin et al). When looking at those TEs aligning perfectly to the reference we obtained 81.73% for Berlin's genome and 64.05% for Solares's genome. The 81.73% (4,335/5,304) is different to the one reported by Berlin et al in the main text (4,984/5,274 = 94.5%), most likely due to different software versions (MUMmer, Blast and Bedtools versions are not specified in Berlin et al. and we used blastn v. 2.2.31+; MUMmer v3.23 and bedtools v2.25.0). Beyond that, among the identified TEs, the percentage aligning perfectly to the genome is ~18% less in Solares's genome in comparison with Berlin's. Such a difference is very likely influenced by at least three aspects: 1) Sequencing technology: PacBio SMRT sequencing in Berlin et al vs. Nanopore (ONT) sequencing in Solares et al. 2) Read length and coverage: 90X & mean read length = 9,317bp in Berlin et al. vs. 30.2X & mean read length = 7,122 bp in Solares et al) Assembly strategy: MinHash Alignment Process (MHAP) strategy integrated into the Celera Assembler in Berlin et al. vs Canu Assembler followed by short-read (Illumina) polishing in Rech et al. (this work). While it is very likely that these aspects influence genome assembly quality and therefore TE annotation, we would like to point out that the 32 genomes reported in our manuscript were sequenced to a higher coverage compared with Solaris et al (82x on average vs. 30.2x).

Although we agree with the reviewer that it is interesting to quantify how differences in assemblies (not only strategies but also the quality and coverage of reads) and differences in the pipelines used to annotate TEs do influence the TE annotation, the main point of our paper is to show that annotations based on libraries that contain the diversity of TEs sequences present in several populations outperforms annotations based mostly on a single genome. Moreover, all our genomes were assembled using the same strategy. Still, we have added a few sentences in our manuscript to acknowledge that TE annotation can be affected both by the assembly and the annotation strategy.

"We also used the pipeline applied in Berlin et al (2015) to annotate TEs in an ISO-1 assembly based on PacBio sequencing, to annotate TEs in our ISO-1 assembly based on the Solares et al (2018) ONT reads. We found that 18% more TE insertions were annotated when using the Berlin et al (2015) assembly, suggesting that besides TE annotation pipelines, sequencing and assembly strategies can also influence the annotation of TEs in genomes."

2) In the Introduction, the authors state that they used strains from 5 different climatic regions. However, in the first paragraph of the Results section, they say 24 strains were collected from 11 geographic locations. For consistency and to avoid confusion, it might be best to explain the geographic regions of origin of the strains initially and be consistent about the numbers throughout the manuscript. For example, in the “A new manually curated....” section, the authors mention 12 geographic regions (is that 11 + North Carolina strains ?).

>>>The information can be found in the same sentence the reviewer mentions. And in the next sentence we make clear that there are 12 populations in total.

“Most of these strains —24 out of 32— were collected from 11 geographical locations across Europe, while the other eight strains were originally taken from Raleigh, North Carolina, USA (Huang et al 2014). These 12 populations represent five different climatic regions belonging to three main climatic types: arid, temperate, and cold (Figure 1; Table S1).”

We have modified these sentence to make it even more clear:

“These 32 strains were collected from 12 geographical locations: 24 strains were collected from 11 European locations and eight strains were collected in a North American populations (Huang et al 2014). These 12 populations represent five different climatic regions belonging to three main climatic types: arid, temperate, and cold (Figure 1; Table S1).”

3) The TE names in the X axis of Fig. 2A were difficult to read. I wonder if the font size could be increased a tad more ?

>>> Thanks for pointing this out to us. We have now increased the font size in this figure.